# Current Opportunities for Targeting Dysregulated Neurodevelopmental Signaling Pathways in Glioblastoma

**DOI:** 10.3390/cells11162530

**Published:** 2022-08-15

**Authors:** Danijela Drakulic, Marija Schwirtlich, Isidora Petrovic, Marija Mojsin, Milena Milivojevic, Natasa Kovacevic-Grujicic, Milena Stevanovic

**Affiliations:** 1Laboratory for Human Molecular Genetics, Institute of Molecular Genetics and Genetic Engineering, University of Belgrade, 11042 Belgrade, Serbia; 2Faculty of Biology, University of Belgrade, 11158 Belgrade, Serbia; 3Serbian Academy of Sciences and Arts, 11000 Belgrade, Serbia

**Keywords:** glioblastoma, GBM subtypes, SHH signaling, Wnt/β-catenin signaling, Notch signaling, TGFβ signaling, BMP signaling, Hippo signaling, RA signaling

## Abstract

Glioblastoma (GBM) is the most common and highly lethal type of brain tumor, with poor survival despite advances in understanding its complexity. After current standard therapeutic treatment, including tumor resection, radiotherapy and concomitant chemotherapy with temozolomide, the median overall survival of patients with this type of tumor is less than 15 months. Thus, there is an urgent need for new insights into GBM molecular characteristics and progress in targeted therapy in order to improve clinical outcomes. The literature data revealed that a number of different signaling pathways are dysregulated in GBM. In this review, we intended to summarize and discuss current literature data and therapeutic modalities focused on targeting dysregulated signaling pathways in GBM. A better understanding of opportunities for targeting signaling pathways that influences malignant behavior of GBM cells might open the way for the development of novel GBM-targeted therapies.

## 1. Glioblastoma

Glioblastoma (GBM), a grade IV glioma [1], represents the most aggressive and the deadliest malignant brain tumor, characterized by fast spreading, infiltrative growth and high level of molecular/cytological heterogeneity [2,3,4,5]. Despite decades of research, patients have poor clinical prognosis, with less than 5% surviving 5 years after diagnosis and a median survival time of less than 15 months [6]. The standard treatment for these tumors is radical surgical resection followed by radiotherapy and chemotherapy using temozolomide (TMZ) as currently the main chemotherapeutic agent [7,8,9]. Unfortunately, rapid recurrence after therapy is detected in virtually all patients, which worsens the prognosis (reviewed in [10]). Major contributors of the aggressive course of the disease include the highly mutated genome of GBM and dysregulated signaling pathways involved in cell proliferation, growth, and survival (reviewed in [11,12,13]).

GBM consists of heterogeneous populations of cells in different phases of differentiation (reviewed in [4,5,14,15]). At the apex of GBM cellular hierarchy is a population of undifferentiated, self-renewing, and highly proliferative stem cells named glioblastoma stem cells (GSCs) (reviewed in [16]). These cells are characterized by high plasticity and the ability to give rise to heterogeneous cancer cells within the GBM (reviewed in [17,18]). Multiple studies demonstrated that, besides heterogeneity, GSCs play an indispensable role in the formation, growth, and progression as well as in therapeutic resistance and recurrence of GBM [19,20,21], indicating that these cells could be a crucial target for treatment (reviewed in [18,22]).

Our knowledge of the molecular biology of GBM has increased in recent decades. There are several cellular events during gliomagenesis. They include genetic instability, loss of cell cycle control, overexpression of growth factors and their receptors, angiogenesis, migration, invasion, and aberrant apoptosis (reviewed in [23]). All these processes are regulated by various signaling pathways, and accordingly, targeting key molecules in signaling pathways holds promise for developing novel therapeutic approaches. Targeting of key signaling pathways found deregulated in GBM, such as EGFR, PDGFR, PI3K–PTEN-Akt-mTOR, cell cycle-associated pathways (CDK4/6, CDKN2A/B), P53, pRB, RAS/MAPK, and STAT3, has been previously described in detail (reviewed in [11,24,25,26]).

Strong evidence in the last decade has suggested that GBMs may actually arise from neural stem cells (NSCs) residing in the lining of lateral ventricles that undergo malignant transformation (reviewed in [27,28]). Furthermore, the same studies underscored the parallels between neural development and gliomagenesis [27,28]. A comparison of the lineage hierarchy of the developing human brain to the transcriptome of GBM cells and GSCs derived from IDH-mutant astrocytoma, WHO grade 4, revealed that this type of brain tumor develops along neurodevelopmental gene programs encompassing a rapidly dividing progenitor population [29]. IDH-mutant astrocytoma, WHO grade 4, is hierarchically organized into three cell lineages that correspond to three normal neural lineages, astrocytic, neuronal, and oligodendrocytic, with progenitor cancer cells at its apex [29]. The results of genetic studies from more than 500 GBMs indicated that during tumor growth and invasion, these cells employ the same signaling networks as NSCs during neurogenesis and/or gliogenesis (reviewed in [13]). Signaling pathways involved in the regulation of nervous system development are dysregulated in CNS malignancies including GBM. Having in mind the aforementioned, in the present review, we focus on current strategies for targeting major neurodevelopmental signaling pathways that have been described to promote GBM growth and invasion (reviewed in [13,30]). These signaling pathways include Sonic Hedgehog (SHH), Bone Morphogenetic Protein (BMPs), Transforming Growth Factor β (TGFβ), Wnt, Notch, Hippo and retinoic acid (RA) pathway (reviewed in [30]). Further, these signaling pathways are active in GSCs, a subpopulation of cells within the tumor responsible for increased resistance to chemotherapy and radiotherapy [31,32]. In GSCs, these signaling pathways are mainly responsible for regulation of self-renewal or differentiation [31,32,33,34,35]. In addition to their role in the maintenance of GSCs, they play important roles in the regulation of proliferative, migratory and invasive abilities of GBM cells, contributing to the growth and progression of GBM.

Having in mind the inter- and intra-heterogeneity of GBMs (reviewed in [14]), stratifications of GBMs into molecular subtypes, including profiling of signaling pathway status, is important for clinical management of patients with GBMs.

## 2. Molecular GBM Subtypes

Many different classifications of GBM into subtypes have emerged over the years. Through these classifications, different GBM subtypes are characterized by different features and gene alterations, and they should be considered for different targeted treatments for fighting GBM (Figure 1).

One of the first molecular stratifications of GBM is based on the presence/absence of mutations in isocitrate dehydrogenase 1 (*IDH1*) or 2 (*IDH2*) genes (Figure 1, Appendix A). Based on this, GBM is subdivided into two subtypes: *IDH* wild-type GBM and *IDH* mutant GBM (reviewed in [36]). *IDH* wild-type GBM is found in more than 90% of all patients (mostly in elderly patients), and the vast majority of *IDH* wild-type GBM are primary GBM that develop de novo (reviewed in [36]). *IDH* mutant GBM is detected in less than 10% of patients with GBM (almost all in young adults) and the vast majority of *IDH* mutant GBM represent secondary GBM, developed from diffuse or anaplastic astrocytoma (reviewed in [36]). Based on the progress in the diagnosis and management of gliomas from 2016, the term *IDH*-mutant GBM is discontinued and replaced with *IDH*-mutant astrocytoma, WHO grade 4 (Figure 1, Appendix A) [37].

Based on gene expression profile, Verhaak and co-workers classified GBM into four subtypes: proneural, neural, classical, and mesenchymal [38] (Figure 1, Appendix A). These subtypes differ in mutations, genes expression, as well as responses to chemo- and radiotherapy (reviewed in [39]). Results obtained by Wang and co-workers, analyzing *IDH* wild-type GBMs, indicated neural subtype as normal neural lineage contamination [40]. Previously, neural subtype has been related to the margin of the tumor where normal neural tissue is likely to be detected [41,42]. The mesenchymal subtype is associated with poor overall survival and poor response to radiotherapy, while the proneural is related to a more favorable outcome [43,44,45,46]. Furthermore, it was detected that a GBM tumor from one patient can harbor cells having characteristics of different GBM molecular subtypes [47]. Additionally, switching from one GBM subtype to another within one GBM tumor has been demonstrated (reviewed in [39]).

Ensenyat-Mendez and co-workers constructed the iGlioSub classifier using machine learning, computational biology algorithms and prioritization of highly informative transcriptomic and epigenomic features which, based on gene expression and DNA methylation profiles, classified GBMs into classical, mesenchymal and proneural subtypes (Figure 1, Appendix A). This classifier showed better performance in stratifying patients compared to stratification based on gene expression profiles only [48].

An integrative approach applied by Neftel and co-workers revealed that tumor cells in GBM exist in four cellular states that recapitulate neural progenitor-like (NPC-like), oligodendrocyte-progenitor-like (OPC-like), astrocyte-like (AC-like) and mesenchymal-like (MES-like) states (Figure 1, Appendix A). MES-like cells are further divided into hypoxia-independent (MES1) and hypoxia-dependent (MES2) sub-groups, while NPC-like cells are divided into NPC1 and NPC2 sub-groups that were distinguished by inclusion of OPC-related genes in NPC1 and neuronal lineage genes in NPC2 (Figure 1, Appendix A) [49].

Three stratifications of GBM into subtypes are proposed based on DNA methylation profiles (Figure 1). In stratification I, Ma and co-workers identified three different GBM prognosis subgroups (Cluster 1–3) with Cluster 3 associated with poor prognosis and relatively lower sample methylation level, and Cluster 2 associated with the best prognosis [50]. On the other side, Brennan and co-workers stratified GBM patients into six methylation classes (M1, M2, M3, M4, M5 (G-CIMP), M6) [51]. Based on DNA methylation in CNS tumors, Capper and co-workers grouped most of the IDH-mutant astrocytoma, WHO grade 4, into methylation subclass “A IDH, HG”, while *IDH* wild-type GBM were stratified into seven methylation classes (“GBM, G34“, “GBM, RTK I”, “GBM, RTK II”, “GBM, RTK III”, “GBM, MES”, “GBM, MID”, “GBM, MYCN”) [52].

Based on multi-omics data (CNVs, gene expression, protein and phosphoprotein abundances), Wang and co-workers defined three clusters in *IDH* wild-type GBMs-nmf1 (proneural-like), nmf2 (mesenchymal-like) and nmf3 (classical-like) (Figure 1) enriched in neuron activity-related pathways, immune response pathways and cell cycle pathways, respectively [53]. Furthermore, classification of GBM into lineage-specific subtypes which possess diverse functional properties and therapeutic vulnerabilities was described (Figure 1, Appendix A) [54]. Based on the results of single-cell RNA sequencing (scRNA-seq) of GSCs and malignant cells from primary GBM tumors, Richards and co-workers stratified cells along a transcriptional gradient from a ‘Developmental’ state to an ‘Injury Response’ state [55,56]. On the other hand, Garofano and co-workers identified four tumor cell states in GBM (proliferative/progenitor, neuronal, mitochondrial and glycolytic/plurimetabolic) converging on two biological axes-neurodevelopmental and metabolic (Figure 1) [57].

Since long noncoding RNAs (lncRNAs) have important roles in development and progression of GBM, and tumor-associated immune processes, Yu and co-workers used immune-related lncRNAs signature for classification of GBM into four subtypes (A-D) (Figure 1), with GBM subtype A showing the most favorable prognosis [58]. Additionally, based on immunohistochemical results in combination with DNA copy number and DNA methylation profiles, Motomura and co-workers stratified GBM into four subtypes: oligodendrocyte precursor type (OPC), differentiated oligodendrocyte type (DOC), astrocytic mesenchymal type (AsMes) and mixed type (Figure 1, Appendix A) with OPC type showing the most favorable outcome [59]. Based on growth factors and cytokines expression profiles of GBM patients, Hu and co-workers identified three GBM subtypes (type I-III) (Figure 1, Appendix A) [60]. Prognosis is poorer for patients with GBM Type III compared to patients with Types I and II [60].

According to gene expression levels and overall survival of patients, three prognostic GBM subtypes were identified: invasive (poor), mitotic (favorable), and intermediate (Figure 1) [61].

Results of microarray analysis revealed the presence of two subtypes of GSCs, proneural and mesenchymal [43,62]. These two subtypes are phenotypically different; mesenchymal GSCs are more invasive, angiogenic and more resistant to radiotherapy compared to the proneural subtype [43,62]. Furthermore, a proneural subtype of GSCs might be switched to the mesenchymal upon certain conditions, such as radiation treatment [62], anti-angiogenic therapy [63], increased intracellular levels of reactive oxygen [64], upregulation of transglutaminase 2 (TGM2) [65], upregulation of N-Myc downstream regulated gene 1 (*NDRG1*) [66], activation of *NFE2L2* transcriptional network [64], upregulation of TAZ expression [67], and activation of nuclear factor kappa B (NF-κB) [43].

Stratification of GBMs into different molecular subtypes enables the identification of markers significant for diagnosis, prognosis, and more effective treatment. Precise identification of GBM molecular subtypes is important for improving clinical management of GBM and might lead to targeted molecular therapy for GBM patients.

Molecular GBM stratification revealed that multiple signaling pathways are dysregulated in GBM. Thus, the development of therapies that are focused on targeting dysregulated signaling pathways in GBM provides a new avenue for improving the clinical management of GBM patients. Among others, dysregulated signaling pathways in GBM include SHH, Wnt/β-catenin, Notch, BMP, TGFβ, Hippo and RA signaling pathways. In this review, we have summarized the current data concerning the approaches for targeting these signaling pathways in GBM.

## 3. Targeting Sonic Hedgehog Signaling Pathway in GBM

The Hedgehog (HH) signaling is a well-conserved pathway in animals that plays a pivotal role during embryonic development, tissue homeostasis, maintenance of adult stem cells and regeneration [68,69,70]. Its role is critical for the development of brain and spinal cord including midbrain and ventral forebrain neuronal differentiation and proliferation [71,72,73]. There are three mammalian HH ligand proteins, Sonic Hedgehog (SHH), Indian Hedgehog (IHH), and Desert Hedgehog (DHH). Canonical HH signaling occurs by binding HH ligand protein to a transmembrane receptor protein patched (PTCH) [74]. In the absence of HH ligands, PTCH functions as an inhibitor of another transmembrane protein smoothened (SMO). Binding of HH ligands to PTCH receptor relieves the suppression of SMO, resulting in downstream activation of final effectors, GLI transcription factors (GLI1, GLI2, and GLI3) responsible for the transmission of HH signaling to downstream target genes (Figure 2) [75,76].

Accumulating evidence suggests that aberrant activation of the HH signaling pathway by deregulation of any of its components may be involved in the development and progression of cancers and diseases. Accordingly, dysregulation of SHH signaling has been implicated in the initiation and/or maintenance of different tumor types including GBM [77,78]. The transcriptomics data on 149 clinical cases of The Cancer Genome Atlas-Glioblastoma database showed a strong correlation between *PTCH1* and *GLI1* upregulated expression in GBM indicating that activation of the canonical SHH pathway might be associated with this malignancy [79]. Furthermore, it was demonstrated that the endothelial cells within the tumor microenvironment (TME) provide the SHH ligand that activates the HH signaling pathway in GBM cells and promotes the appearance of GSCs, as demonstrated by increases in tumorigenicity and expression of stemness genes [80]. It has been revealed that GSCs highly express SHH and that targeting this signaling pathway may overcome chemoresistance and provide a novel therapeutic strategy [22,80,81].

Currently, there are two approaches to inhibiting SHH signaling: ligand-dependent approach, by antagonizing the SMO receptor, and ligand-independent approach, by inhibiting its final effectors, GLI transcription factors [77]. In preclinical studies, various SMO inhibitors exerted antiproliferative effect against tumor cells including those originating from GBM. Cyclopamine, isolated from *Veratrum californicum* plant, is a natural inhibitor of SMO that consequently blocks the SHH signaling pathway [77,82]. Carballo and co-workers presented data showing that SHH pathway inhibition with cyclopamine interferes with GBM cell viability and also suggested that cyclopamine inhibition of the SHH pathway prior to TMZ treatment could reduce the aggressiveness of the tumor cells by sensitizing the GSCs to TMZ [83]. Among SMO inhibitors, there are two inhibitors, vismodegib, and sonidegib, approved by the FDA for the treatment of locally advanced and metastatic basocellular carcinoma [84]. Bureta and co-workers presented data on the antiproliferative effect of vismodegib on GBM cell lines alone or in combination with arsenic trioxide (ATO) and TMZ [85].

Presently, there are several ongoing clinical trials evaluating SMO inhibitors for the treatment of different types of brain tumors. Initially, SMO was the principal target for the development of SHH inhibitors, and the first clinical trial performed using an SMO inhibitor to treat a brain tumor was conducted in 2008 in a male patient with metastatic medulloblastoma who was treated with a novel HH pathway inhibitor, GDC-0449 (vismodegib) [86]. Treatment led to rapid regression of the tumor and reduction in symptoms, but only transiently [86]. It was observed that the HH pathway inhibition resulted in an incomplete response that led to tumor re-growth, and unfortunately, the patient died after five months of treatment [86]. Vismodegib was evaluated in clinical trials against GBM, but so far, it has not demonstrated clinical benefit as a single agent. Patients with recurrent GBM, in a phase 2 trial, were randomized to a pre-operative and post-operative vismodegib group (group I) versus only the post-operative vismodegib group (group II), with the idea that HH pathway inhibitors selectively target GSCs. Although a significant decrease in the number of CD133+ neurospheres was observed in group I, vismodegib was not efficacious as a single agent in recurrent GBM [87]. It is important to highlight that the efficiency of vismodegib is still being evaluated in one clinical trial for medulloblastoma (NCT01878617) and one for GBM (NCT03158389). Another FDA-approved SMO inhibitor sonidegib was also evaluated in relapsed medulloblastoma in a phase 1/2 trial, with both adult and pediatric patients. Fifty percent of the patients with activated HH pathway in their tumor had a response to sonidegib, which was translated to longer disease-free survival [88]. Currently, two additional SMO inhibitors are under evaluation in clinical trials for the treatment of GBM, and they both belong to drugs approved for other diseases (NCT03466450; NCT02770378). The first is glasdegib, a small molecule inhibitor of SMO and FDA-approved medication for acute myeloid leukemia [89]. The other one is itraconazole, an antifungal drug used for the treatment of fungal infections, which is currently in phase I clinical trial in combination with TMZ (CUSP9v3 Treatment Protocol) for recurrent GBM. Previously, itraconazole has been explored as an anticancer agent for patients with basal cell carcinoma, non-small cell lung cancer, and prostate cancer [90,91,92,93].

Most HH inhibitors that have entered clinical trials targeted SMO, although several mechanisms of resistance to SMO inhibitors have been identified. This is why alternative SHH antagonists that act directly on GLI transcription factors are already being tested in the brain and central nervous system tumors as co-adjuvant therapy with TMZ (reviewed in [77]). ATO is a drug that is being tested for gliomas in phase I and II clinical trials. ATO is an FDA-approved drug which was first used for the treatment of patients with acute promyelocytic leukemia. It has been shown that ATO inhibits GLI-dependent growth in a medulloblastoma mouse model [94]. A recent study demonstrated that treatment of patients in combination with ATO, TMZ, and radiation apparently does not improve the overall outcome in GBM patients [95]. Genistein is an isoflavone found in legumes which is able to inhibit GLI1, and there was a clinical study of its potential against brain malignancies (NCT02624388); unfortunately, the study was terminated due to poor enrolment. However, clinical trials of genistein in cancer therapy have been conducted for other malignancies, including breast, bladder, and prostate cancer. Currently, its clinical applications are still limited due to its poor solubility and bioavailability [96]. Honorato and co-workers reported that selective ligand-independent inhibition of SHH by GANT-61 through targeting GLI1 increased the oxidative stress damage potentiating TMZ effect and inducing cell death in GBM cell lines [97]. Another preclinical study also showed that GANT61 sensitizes glioma cells to TMZ treatment [98]. However, currently, there are no data regarding its clinical evaluation.

Nowadays, the SMO receptor is the primary target used for the development of SHH pathway inhibitors, and there are several ongoing clinical trials for different types of brain tumors. On the other hand, several reports demonstrated that SHH could signal through a noncanonical route, and while it is still unclear how the cells select between canonical and non-canonical routes, it is understood that efficient targeting of downstream effectors (GLIs) could lead to promising results in clinical trials. In conclusion, it is considered that the best way to control the tumor recurrence is to evaluate and establish novel protocols combining SHH signaling inhibition with conventional therapies.

## 4. Targeting Canonical Wnt/β-Catenin Signaling Pathway in GBM

Wnt/β-catenin signaling pathway plays essential roles in embryonic development and the maintenance of homeostasis and regeneration of adult tissues [99,100]. Canonical Wnt/β-catenin signaling pathway is the most studied part of the complex and evolutionarily conserved Wnt signaling [101].

β-catenin is a central player in the canonical Wnt signaling pathway [102,103]. When Wnt signaling is inactive, the level of β-catenin is kept low due to the activity of the destruction complex in the cytoplasm [104]. The complex is consisting of APC (adenomatous polyposis coli), AXIN1/2 (axis inhibition proteins 1/2), CK1 (casein kinase 1), and GSK3β (glycogen synthase kinase 3β) [104]. Two scaffold proteins, APC and AXIN, bring β-catenin in the position for CK1/GSK3β-mediated phosphorylation which prime it for subsequent ubiquitination and proteasomal degradation [104,105]. Decreased concentration of β-catenin in a cytoplasm prevents its nuclear translocation. In the absence of β-catenin, members of TCF/LEF (T-cell factor/lymphoid enhancer-binding factor) families of transcription factors remain in complexes with corepressors, the Groucho and TLE (transducin-like enhancer), bound to Wnt responsive elements thus repressing transcription of Wnt target genes [106].

Activation of canonical Wnt signaling starts by binding of WNT ligands (family of 19 secreted glycoproteins in mammals) to the Fzd seven-transmembrane receptor (Frizzled) and LRP5/6 (co-receptor low-density lipoprotein receptor-related protein 5, 6) [107,108]. Upon activation, Fzd interacts with Dsh (Disheveled) cytoplasmic protein. Dsh also interacts with Axin, and LRP5/6, via Axin interactions with the coreceptor. These interactions inactivate β-catenin destruction complex [107,108]. Increased concentration of stabilized β-catenin in cytoplasm leads to its nuclear translocation. In the nucleus, β-catenin forms complexes with the members of the TCF/LEF family of transcription factors and co-activators, e.g., CBP (CREB-binding protein) and p300, and enhances transcription of Wnt target genes [106].

Aberrant activity of Wnt/β-catenin signaling is associated with diverse human diseases including cancer [99,100]. Aberrant activation of the Wnt/β-catenin signaling pathway is involved in GBM pathogenesis and progression, maintaining of GSC stemness and acquisition of radio- and chemotherapy resistance [15,109,110,111].

Mutations in key components of Wnt signaling e.g., APC, β-catenin, and AXIN are not hallmarks of GBM, although several studies linked mutations in Wnt signaling proteins to gliomagenesis. In a study on seven patients, Tang and co-workers showed that *APC* gene mutations occurred in 13% of cases, with a mutation frequency of 14.5% [112]. A study by Morris correlated homozygous deletion in tumor suppressor *FAT1* (atypical cadherin 1), negative regulator of the Wnt pathway, with the activation of Wnt signaling. Homozygous deletion of *FAT1* occurred in 57% of GBM and was associated with a prolonged survival [113,114]. A study by Sareddy and co-workers showed overexpression of PELP1 (proline-, glutamic acid-, and leucine-rich protein 1) in 100% of the GBM samples. PELP1 is co-regulator of several nuclear receptors, and a potent activator of β-catenin and Wnt signaling [115]. MicroRNA and lncRNA profiling of GBM versus the normal brain revealed the role of epigenetic mechanisms in the activation of Wnt signaling in GBM [116,117].

In search of anticancer therapeutics, numerous natural and synthetic antagonists and agonists were isolated or designed to target major cascades in Wnt/β-catenin signaling pathway, i.e., inhibitors targeting Wnt ligand/FZD receptor complex, inhibitors and agonists targeting the β-catenin-destruction complex and inhibitors targeting β-catenin/TCF complex [108]. Although the Wnt pathway is a validated target in cancer, and drugs targeting this pathway in various cancer types have entered the clinical trials, there are still no FDA-approved drugs targeting this pathway [118]. Unfortunately, only a few candidates have reached early phase clinical trials as therapies targeting GBM [24,119].

Extensive search and chemical screening for Wnt inhibitors identified numerous small molecules and biologics that targeted the Wnt signaling cascade [120]. Small molecules that act as Wnt agonists or antagonists, modulation of Wnt signaling activity induced by these molecules, and their effects on the properties of GBM cells and GSCs are listed in Table 1, Section A [121,122,123,124,125,126,127,128,129]. A schematic representation of their action on the Wnt signaling cascade is given in Figure 3.

**Figure 3 cells-11-02530-f003:**
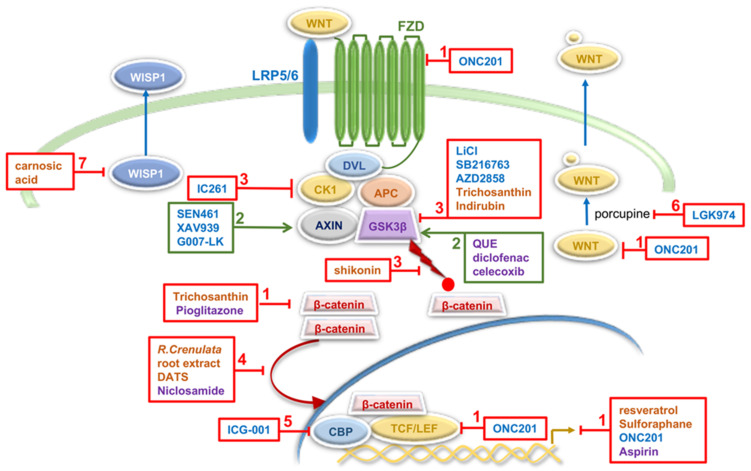
Overview of the Wnt/β-catenin signaling cascade, targets for potential therapeutic intervention in GBM and molecules investigated in preclinical and clinical studies (based on the data listed in Table 1). Mechanisms of modulation of Wnt/β-catenin activity in GBM (described in detail in Table 1) include (1) down-regulation of expression of Wnt components and Wnt targets, (2) promotion of β-catenin degradation, (3) increasing β-catenin stability, (4) inhibition of β-catenin nuclear translocation, (5) inhibition of β-catenin/CBP transcription complex, (6) down-regulation of WNT secretion and (7) inhibition of Wnt/β-catenin-WISP1 signaling. Small molecules are presented in blue letters, natural agents in orange and repurposed drugs in purple. Molecules in red boxes inhibit expression and activity of denoted Wnt component or processes within the Wnt cascade. Molecules in green increase activity of denoted components. Red circle represents phosphate group; yellow circle represents palmitoyl groups. Based on [120,130] and references included in the main text and the Table 1.

**Table 1 cells-11-02530-t001:** Molecules targeting Wnt/β-catenin cascade, mechanisms of action and effects on GBM cells and GSCs: A: Small molecules; B: Natural agents; and C: Repurposed drugs.

Molecule	Modulation of Wnt Signaling Activity	Effects on GBM Cells and GSCs Properties	Reference
A: Small molecules			
ONC201	inhibits expression of components of Wnt pathway and Wnt targets	induces apoptosis in GBM cells induces cytotoxicity in chemo- and radiation-resistant GBM patient samples inhibits the growth of GSCs in 3D neurospheres established from human GBM tumors inhibits tumor growth in GBM mouse models	[121,127]
SEN461	induces AXIN stabilization	inhibits anchorage-independent growth of human GBM cell lines and patient-derived primary tumor cells reduces tumor growth in a mouse GBM xenograft model	[122]
XAV939	antagonist of tankyrase-enzymes involved in the degradation of AXIN	decreases the survival and clonogenicity of GBM cells reduces GSC population increases radiosensitization in in vivo radiation model derived from a single human GBM specimen	[125]
LiCl SB216763	inhibits GSK3β	induces the expression of differentiation markers in GBM cells depletes GSCs population reduces colony formation and induces cell death in GBM cell-lines	[126]
G007-LK	inhibitor of tankyrase-enzymes involved in the degradation of AXIN	decreases in vitro proliferation and sphere formation in primary GSC cultures reduction in GSC sphere formation in cotreatment with TMZ	[124]
IC261	inhibitor of CK1	inhibits growth of GBM cells and GSCs in vitro and induces growth inhibition of human GBM xenografts in mice	[129]
LGK974	inhibitor of porcupine proteins that modulate Wnt ligands	acts synergistically with TMZ to inhibit growth of GBM cells	[128]
ICG-001	CBP antagonist	reduces proliferation and survival of GBM cells	[123]
AZD2858	inhibits GSK-3β	reduces proliferation and survival of GBM cells inhibits the invasion and migration of GBM cells	[123]
B: Natural agents			
shikonin	inhibits β-catenin phosphorylation	inhibits proliferation, migration and invasion of GBM cells	[131]
Trichosanthin	inhibits expression of Wnt components	inhibits proliferation, invasion and migration and induces apoptosis of GBM cells	[132]
*R. crenulata* root extract	decreases nuclear localization of β-catenin	inhibits proliferation and tumorsphere formation and promotes differentiation of GBM cells	[133]
resveratrol	decreases expression of Wnt signaling components and Wnt targets	inhibits proliferation, motility and invasion of GSCs	[134]
carnosic acid	decreases expression of WISP1	reduces GSC viability suppresses GSC tumorsphere formation inhibits the growth of GSC-derived xenografts	[135]
Indirubin	inhibitor of GSK-3β	reduces invasion of GBM and GSC-enriched neurospheres both in vitro and in vivo improves survival of intracranial glioma-bearing mice	[136]
DATS	decreases nuclear β-catenin level	inhibits cell growth, induces apoptosis and decreases migration and invasion in GBM cells	[137]
Sulforaphane	inhibits Wnt/β-catenin signaling	enhances TMZ-induced apoptosis	[138]
C: Repurposed drugs			
NSAIDs diclofenac celecoxib aspirin	reduces phosphorylation of GSK3β inhibits expression of Wnt targets	inhibits proliferation, colony formation and migration of GBM cells inhibits proliferation and invasion and induces apoptosis of GBM cells	[139] [140]
Niclosamide	decreases concentration of β-catenin in the nucleus	decreases cell viability, exerts antimigratory effects and inhibits the malignant potential of primary GBMs combined treatment with TMZ inhibits viability, stemness, and invasive properties of human GBM tumorspheres and decreases tumor growth in mouse xenograft models	[141] [142]
QUE	decreases phosphorylation of GSK3β	suppresses GSCs-initiated tumor growth in mouse models of gliomas acts synergistically with TMZ to suppress growth of TMZ-resistant tumors originated from GSCs	[143]
Pioglitazone	inhibits β-catenin expression	reduces cell viability, suppresses invasion and induces apoptosis of GBM cells induces decrease in cell viability and proliferation of GSC lines isolated from GBM patients	[144] [145]

LiCl, lithium chloride; DATS, garlic-derived diallyl trisulfide; NSAID, nonsteroidal anti-inflammatory drugs; QUE, Quetiapine.

The most promising agent is ONC201, a member of the imipridone class of small molecules. This dopamine receptor D2 antagonist is a widely studied inhibitor of GSC markers expression and suppressor of signaling pathways associated with GSC self-renewal, glioma-initiation and progression, and therapy resistance including Wnt ligands, receptors, and effectors (WNT16, FZD2, FZD4, TCF7L2) [127,146,147]. Additionally, ONC201 showed promising results in phase II clinical trial in patients with recurrent GBM [148]. ONC201 is currently in phase II clinical trial for patients with recurrent GBM and distinct subtypes of glial tumors H3 K27M mutant and Midline Glioma (NCT02525692).

Several biologics targeting Wnt signaling were investigated in various cancer types [120]. Fusion protein Ipafricept (OMP54F28; IPA), which competes with WNT ligands for binding to FZD8 receptor [149,150,151], and Vantictumab (OMP-18R5), a monoclonal antibody targeting FZD1, FZD2, FZD5, FZD7, and FZD8 receptors [150,151], act as Wnt/FZD antagonists and reached clinical I/II phase for breast, pancreatic, hepatocellular and ovarian cancer [120]. However, there are no available data about ongoing clinical trials or plans for further investigations of these biologics in GBM treatments.

Numerous natural products act as inhibitors of the canonical Wnt signaling and show significant therapeutic relevance in various cancer model systems [130]. They mainly act via β-catenin by regulating different steps involved in its stability and transcriptional activity, i.e., expression of β-catenin and its phosphorylation, degradation and nuclear translocation [130]. It is important to point out that about 33% of FDA-approved anticancer drugs are natural products and their derivatives [152] and that natural products represent a major source for drug discovery and development [153,154]. Natural agents acting through Wnt signaling in GBM are presented in Table 1, Section B [131,132,133,134,135,136,137,138]. They affect different processes in the Wnt cascade: β-catenin phosphorylation (shikonin), expression of key proteins in the Wnt/β-catenin signaling pathway (trichosanthin), nuclear translocation of β-catenin (*R. crenulata* root extract, DATS), expression of Wnt signaling components and Wnt targets (resveratrol, sulforaphane), activity of GSK-3β (indirubin), and Wnt/β-catenin-WISP1 signaling (carnosic acid) (Table 1, Section B and Figure 3) [131,132,133,134,135,136,137,138]. WISP1 (Wnt-induced signaling protein 1) is secreted by GSCs to facilitate a pro-tumor microenvironment and promote survival of both GSCs and tumor-associated macrophages [135].

One of the promising strategies to develop Wnt-targeting anti-GBM therapies is drug repurposing. Several studies revealed tumor-suppressive effects of various drug types: nonsteroidal anti-inflammatory drugs (NSAIDs); niclosamide, an anthelmintic drug for treating intestinal parasite infections; quetiapine (QUE), an atypical antipsychotic drug and pioglitazone, an anti-diabetic drug used to treat type 2 diabetes (Table 1, Section C and Figure 3) [139,140,141,142,143,144,145]. The most promising repurposed drug targeting Wnt signaling in GBM is celecoxib. This NSAID was tested in GBM patients in several phase I and II clinical trials in combination with other drugs. Some trials are still ongoing but preliminary results are not encouraging [24,155]. Tested in phase II trial in newly diagnosed GBMs, celecoxib had no survival benefit when combined with TMZ (NCT00112502) [24].

Several studies point to the translational potential of therapeutic approaches that combined Wnt pathway inhibitors and chemotherapy. For example, Huang and co-workers discovered that canonical Wnt signaling plays a crucial role in stemness activation and chemoresistance in GBM-associated vascular endothelial cells through the HGF/c-Met/β-catenin axis [156]. In endothelial cells under glioma conditions, HGF (hepatocyte growth factor) induces direct β-catenin phosphorylation at Ser675 by HGF receptor kinase c-Met, leading to the β-catenin nuclear translocation and LEF1-mediated activation of Wnt target genes [156]. RNA seq data revealed activation of genes associated with stemness that govern endothelial cells transformation into mesenchymal stem cell-like cells. In addition, they detected increased expression of ABCC1 (multidrug resistance-associated protein 1 (MRP-1)), which is responsible for drug efflux and chemoresistance of these cells. In further investigation of the potential therapeutic relevance of these findings, the authors examined the effects of treatment with Wnt inhibitor XAV939 and TMZ. The combined treatment of XAV939 and TMZ extended mouse survival and inhibited tumor growth, suggesting that Wnt inhibition with XAV939 sensitizes GBM to TMZ chemotherapy [156]. The authors also revealed that Wnt-mediated chemoresistance is potentially endothelial cells-selective mechanism in GBM [156].

In addition, canonical Wnt signaling is involved in one of the molecular pathways of the GBM cells vessel co-option [157], a mechanism of the tumor cells movement towards and along the pre-existing vasculature [158]. Griveau and co-workers described Olig2-Wnt7 signaling axis responsible for single-cell vessel co-option and consequent increased infiltration of patient-derived oligodendrocyte precursors-like cells with preservation of the blood–brain barrier (BBB) [159]. In contrast, Olig2- and Wnt7-negative astrocyte-like GBM cells show collective clusters vessel co-option, which leads to the destruction of the BBB and consequent inflammation [159,160]. It has been also shown that vessel co-option is invasion strategy for orthotopically implanted mouse, rat and human glioma cells and GSCs [161].

In vivo and ex vivo inhibition of Wnt signaling with porcupine inhibitor LGK974 reduced vessel co-option and improved survival in combination with TMZ treatment [159]. Additionally, in patient-derived proneural cell lines, LGK974 treatment down-regulated OLIG2 and WNT7A expression and up-regulated VEGFA, while treatment with VEGF inhibitor B20 increased expression of OLIG2. In vivo, LGK974 treatment increased VEGF expression, whereas B20 treatment increased Wnt activity [159]. In addition, up-regulation of both Wnt7a and Wnt7b were observed in U87 GBM cells resistant to bevacizumab compared to bevacizumab-sensitive U87 cells [159]. This is in concordance with the clinical studies that showed that vessel co-option occurs in some GBM subtypes intrinsically resistant to anti-angiogenic treatment, but it is also a mechanism of resistance to anti-angiogenic treatments (bevacizumab) in the innately angiogenic tumors [158,162]. The ability of GBM cells to switch between co-opting and angiogenic phenotype has great therapeutic relevance. Computational modeling of the inhibition of both mechanisms suggests that sequential inhibition (vessel co-option inhibition followed by anti-angiogenesis therapy) is more efficient than simultaneous blockage [163]. These findings also highlight the importance of classification of the GBM patients based on vessel co-option occurrence.

Inhibition of Wnt in GBM is also challenging due to the essential roles of this pathway in CNS vascularization and BBB integrity [100]. During CNS development, neural progenitors-derived Wnt7a and Wnt7b ligands activate canonical Wnt signaling in vascular endothelium, thus promoting vascularization of CNS and formation of BBB [164]. The same mechanism maintains integrity of BBB in adult brain. In brain endothelial cells, Wnt7 binds to the co-receptor complex RECK-GPR124, consisting of RECK (reversion-inducing cysteine-rich protein with Kazal motifs) and GPR124 (G protein-coupled receptor 124), to activate canonical Wnt signaling through Fz receptor [165,166]. Martin and co-workers took advantage of the selective binding of linker domain of Wnt7a to Fz receptor only in the presence of RECK-GPR124 complex to restrict Wnt activation to the endothelial cells of CNS and avoid their pleiotropic Fz signaling [167]. The authors engineered the Wnt7a mutant ligand, a Gpr124/Reck agonist that showed no off-target activity in the brain [167]. Delivered in the circulation of mouse model of brain tumor, by adeno-associated virus injection, the Gpr124/Reck agonist restored Wnt signaling in endothelial cells, normalized BBB function and reduced GBM expansion [167].

This approach also showed improved BBB function in cerebral artery occlusion model of stroke [167], which is in concordance with previous studies that highlighted the role of Wnt signaling activation in CNS vascularization during development and in hypoxic brain injury. Chavali and co-workers revealed that crosstalk between oligodendrocyte precursor cells (OPCs) and endothelial cells regulates vascular development in neonatal white matter in a Wnt-dependent manner [168]. The authors showed that in hypoxic brain injury, OPCs expressed the Wnt7A ligand, which resulted in paracrine activation of the canonical Wnt signaling in endothelial cells in their proximity and expression of Wnt targets Apcdd1 and Axin2 [168]. They also showed that loss of Wnt ligand production led to the decreased proliferation of endothelial cells and disrupted angiogenic sprouting [168]. The authors concluded that Wnt activity is essential for the maintenance of white matter integrity and that expression of Wnt7 in OPCs is an indicator of the white matter susceptibility to a hypoxic injury [168].

Other Wnt ligands also control brain vasculature in pathological conditions. Reis and co-workers showed that forced activation of canonical Wnt signaling in GBM endothelia, up-regulated by glioma-derived Wnt1, increased Dll4 (Delta-like 4) expression and induced Notch signaling, leading to an angiogenic blockage and quiescence of endothelial cells [169]. Additionally, transcriptional activity of β-catenin induced expression of PDGF-B (platelet derived growth factor B) and consequent recruitment of mural cells [169]. Both cascades led to the vessel stabilization and normalization of BBB function [169].

Although selective targeting of Wnt cascade in neuroendothelia seemed unachievable due to its pleiotropic mode of action, a study by Martin and co-workers provides a promising strategy for BBB-focused therapy approach in GBM [167].

## 5. Targeting Canonical Notch Signaling Pathway in GBM

Notch signaling is an ancient and evolutionarily conserved signaling pathway that plays a critical role in multiple cellular processes throughout life, including stem cell maintenance, cell fate decisions, cell proliferation and differentiation (reviewed in [170,171]). Accordingly, it was shown to be essential for neural stem cell maintenance and proper control of neurogenesis in both embryonic and adult central nervous system (CNS) [172]. Activation of Notch signaling can inhibit neurogenesis, maintain neural progenitor identity and, in certain settings, promote gliogenesis and drive binary fate choices, leading to various neuronal cell types (reviewed in [173]).

The canonical Notch signaling pathway involves activation of Notch receptor through series of proteolytic cleavages, resulting in translocation of Notch intracellular domain (NICD) to the nucleus, where it activates transcription of target genes through association with DNA-binding proteins and transcriptional co-activators (reviewed in [174,175]). Activation of Notch signaling is accomplished through physical interactions between the Notch receptor of the signal-receiving cell and the membrane-bound ligands of its neighbor, a signal-sending cell. There are four mammalian Notch receptor paralogs (Notch1, Notch2, Notch3 and Notch4), representing large single-pass type I transmembrane proteins that display redundant as well as unique functions [176,177]. Notch receptors in mammals are activated by five type I transmembrane, three Delta-like (Dll1, Dll3 and Dll4) and two Serrate/Jagged (Jag1 and Jag2) [175] ligands. After translation, Notch polypeptide undergoes a series of glycosylations in the endoplasmic reticulum (ER) [178,179] and then translocates into the Golgi apparatus, where it is cleaved by furin-like convertase (S1 cleavage) into a heterodimer in which Notch extracellular domain (NECD) and Notch transmembrane and intracellular domain are non-covalently bonded and transported to the cell membrane (Figure 4) [180,181]. Ligand binding triggers conformational change in the Notch receptor facilitating a second NECD cleavage (S2 cleavage) by ADAM (a disintegrin and metalloprotease) proteases [182,183]. This process detaches NECD from the cell surface leaving the membrane-bound Notch extracellular truncation (NEXT) fragment, which is then cleaved by γ-secretase complex to release the Notch intracellular domain (NICD) (Figure 4) [184,185]. NICD translocates to the nucleus, where it interacts with the DNA-binding protein CSL [186,187,188] and Mastermind-like transcriptional coactivators (MAML) [189]. The stable ternary complex that is formed [190,191,192] further recruits coactivators, such as histone acetyltransferases (CBP/p300) and chromatin remodeling complexes [193,194] to activate transcription of Notch target genes. On the other hand, when NICD is not present, CSL associates with ubiquitous corepressor (Co-R) proteins and histone deacetylases (HDACs) to repress transcription of Notch target genes. The first Notch target genes to be discovered included *Hairy/Enhancer of split* (*HES*) and *HES-related repressor protein* (*HERP*) gene families (also known as *HES-related with YRPW motif*- *HEY*) that encode basic helix–loop–helix transcriptional repressors that play important roles in lineage-commitment decisions [195,196].

Notch signaling pathway is often deregulated in different cancers including gliomas. Various studies have demonstrated increased expression of distinct components of Notch signaling pathway (such as Notch1, Notch2, Dll1, Dll4 and Jag1) and Notch target genes (*Hey1*, *Hey2*, *Hes1*) in glioma cell lines or primary human glioma samples including GBM [197,198,199].

Depending on the context, Notch signaling pathway may exhibit oncogenic or tumor-suppressive functions in glioma (reviewed in [200]). It has been reported that increased Notch activity is correlated with improved patient survival in defined subsets of glioma [201]. Numerous studies support oncogenic function of Notch signaling in brain tumors. Downregulation of Notch1 receptor or its ligands through RNA interference inhibited proliferation and induced apoptosis of a variety of glioma cell lines and prolonged survival in a mouse orthotopic brain tumor model [198]. Activation of the Notch signaling pathway in GBM-derived neurosphere lines having stem cell-like properties is crucial for their growth in vitro and in vivo [202]. Hu and co-workers have shown that, as in the case with neural stem cells, Notch signaling plays a critical role in the maintenance of the patient-derived glioma stem cells by promoting their self-renewal and inhibiting their differentiation [33]. Endothelial cells in GBM actively participate in this process by providing Notch ligands to Notch receptors expressed in GBM cancer stem-like cells, thereby generating a stem cell niche that enables their self-renewal [203]. Notch signaling also plays a critical role in promoting radioresistance of GSCs via activation of the PI3K/Akt pathway [204] as well as in chemoprotection and repopulation of TMZ-treated gliomas [205].

To date, several approaches for targeting Notch signaling in vitro and in vivo have been developed, with two major classes of Notch inhibitors emerging as promising for the clinical development (Figure 4). In the treatment of cancer, the most utilized are γ-secretase inhibitors (GSIs), which block S3 cleavage and the release of the active form of Notch receptor (NICD) by the γ-secretase complex. The other major approach is usage of monoclonal antibodies (mAbs) against specific Notch ligands or receptors to interfere with their interaction or activation. GSIs were originally developed for the treatment of Alzheimer’s disease because they inhibit cleavage of the amyloid precursor protein that leads to generation of neurotoxic amyloid β-protein. Numerous reports provide evidence of the effectiveness of GSIs against GBM alone or in combination with other therapeutic approaches [206]. One of the most widely used GSIs—DAPT (also known as GSI-IX)—augmented the effect of radiation and reduced proliferation and self-renewal of tumor cells as well as proliferation of endothelial cells, thereby hampering the perivascular niche in GBM explants [207]. Other studies have also shown that Notch blockage by GSIs (DAPT and GSI-I) enhance radiosensitivity of CD133+ GSCs [204,208]. One of the important findings was that the subset of GBM-derived tumor-initiating cells sensitive to three structurally distinct GSIs (DAPT, RO4929097 and BMS-708163) are characterized by a signature enriched in proneural genes and high Notch activity [209]. Notch inhibition by GSIs led to reduced proliferation and self-renewal of these responder GBM-derived tumor-initiating cells and their neuronal and astrocytic differentiation. However, clinical application of these findings is not straightforward because of the intratumoral heterogeneity of GBM. Namely, only about 50% of proneural GBMs also have Notch pathway signature [209].

Although numerous in vitro and in vivo preclinical studies of anticancer effect of GSIs in GBM have shown promising results, to date, only two GSIs, RO4929097and MK-0752, have been tested in clinical trials in that respect. Phase 0/I trial was conducted to evaluate the effect of chemo-radiotherapy in combination with RO4929097 in patients with newly diagnosed GBM or anaplastic astrocytoma [210] (NCT01119599). The combination of RO4929097 with TMZ and radiotherapy was well tolerated, and no dose-limiting toxicities were observed. A substantial reduction in proliferation and NICD expression by tumor cells and blood vessels was detected. Treatment with RO4929097 resulted in specific reduction in the CD133+ cancer-initiating cells population in patient-derived tumor explant cultures. There was a modulation of glioma vasculature during RO4929097 treatment. However, in about one-third of the patients, there was tumor recurrence, which might be the result of tumors switching to a Notch-independent angiogenic profile, underscoring the need of targeting multiple signaling pathways simultaneously in gliomas. Another phase I clinical trial has addressed the issue of angiogenesis, where combined effect of RO4929097 and bevacizumab (Vascular endothelial growth factor inhibitor) in patients with recurrent malignant glioma was evaluated [211] (NCT01189240). The combination of RO4929097 and bevacizumab was well tolerated, but definitive conclusions regarding clinical activity of the drug combination could not be made because of the small number of patients enrolled in the study. A phase II trial of RO4929097 for patients with recurrent and progressive GBM (NCT01122901) demonstrated the lack of activity of RO4929097 against recurrent GBM with minimal inhibition of neurosphere formation in fresh tissue samples [212]. Poor clinical activity of RO4929097 could be explained by autoinduction of RO4909097 metabolism, since it increases CYP3A4 activity in vivo, which might result in a decrease in steady-state drug levels [213]. Another selective GSI, MK-0752, has undergone a phase I trial in children with refractory or recurrent CNS malignancies including GBM [214] (NCT00572182). MK-0752 was well tolerated; however, most patients experienced progression of their disease except for two patients: one patient with ependymoma and one patient with GBM that experienced prolonged stable disease. A phase I pharmacologic and pharmacodynamic study of the GSI MK-0752 in adult patients with advanced solid tumors did not show any clinical activity of MK-0752 in extracranial solid tumors, but a modest level of activity of this drug was observed in patients with various types of glioma [215] (NCT00106145). One patient with an anaplastic astrocytoma had a complete response, and ten patients with various gliomas including GBM had prolonged stable disease.

GSIs cause severe intestinal toxicity on account of inhibiting overall Notch pathway [216,217]. Besides processing Notch receptors, γ-secretase complex cleaves a multitude of other membrane proteins affecting other signaling pathways and likely contributing to the GSIs toxicity [218,219]. In order to inhibit Notch signaling pathway more specifically than GSIs, the researchers have been developing antibodies specifically targeting different receptors and ligands of the Notch signaling pathway (Figure 4). There are two classes of antibodies against Notch receptors: (i) antibodies that target Notch negative regulatory region (NRR) preventing the S2 cleavage by ADAMs necessary for activation of receptor; (ii) antibodies that block interactions between Notch receptor and ligand by targeting epidermal growth factor (EGF)-like repeats of the receptor necessary for ligand binding [220,221]. Antibodies targeting Notch1 (Brontictuzumab, OMP-52M51; NCT01778439), Notch2/Notch3 (Tarextumab, OMP-59R5; NCT01277146) and DLL4 (Enoticumab, REGN421, NCT00187159; Demcizumab, OMP-21M18, NCT00744563) have been tested in phase I trials in patients with solid tumors; however, no GBM patients were included.

Antibodies directed at two or three targets/pathways at the same time (bi- or trispecific IgG-like molecules) have been an attractive and promising strategy in anticancer therapy. In preclinical studies, dual-variable domain immunoglobulin targeting simultaneously DLL4 and VEGF (ABT-165, dilpacimab), significantly inhibited tumor growth and decreased functional tumor angiogenesis in U87-MG human GBM xenograft model [222]. Additionally, in combination with TMZ, ABT-165 substantially increased tumor growth delay compared with either monotherapy alone.

Another approach for more specific targeting of Notch in cancer cells involves usage of antibody–drug conjugates (ADC) that target specific antigens (Notch receptor or ligand) highly expressed in tumor cells to deliver a cytotoxic drug (known as the “payload”) to them [223]. In that context, Notch ligand DLL3 represents an attractive therapeutic target, since it is highly and homogenously expressed in *IDH* mutant gliomas and rarely detected in non-tumor brain tissue [224]. In vitro studies have shown that patient-derived *IDH* mutant glioma tumorspheres could be effectively targeted by the anti-DLL3-ADC, rovalpituzumab tesirine (Rova-T) [224]. This first-in-class anti-DLL3-ADC showed promising results in a phase I study of antitumor activity in patients with recurrent small-cell lung carcinoma (SCLC) that exhibit high DLL3 expression (NCT01901653) [225]. However, further work on Rova-T was terminated because later phase III studies showed a lack of survival benefit for patients with SCLC [226,227]. Nevertheless, DLL3 remains an attractive target for the development of new, improved antibody-based biologics.

Another strategy for inhibiting Notch signaling pathway that has emerged over the years is to block the Notch transcriptional complex using dominant-negative form of the MAML coactivator, stapled peptides or small molecules that interfere with protein–protein interactions between the components of the NICD-CSL-MAML transcriptional complex (Figure 4). The lentivirally expressed dominant negative form of Notch coactivator MAML1 (dnMAML1) has been shown to significantly inhibit Notch signaling and reduce GBM cell growth in vitro and in vivo through induction of G0/G1 cell cycle arrest and apoptosis in GBM cell lines [228]. Opačak-Bernardi and co-workers have developed conjugate of dnMAML peptide, thermo-responsive elastin-like polypeptide (for targeted delivery to tumor cells) and cell penetrating peptide (for enhanced cellular uptake and BBB penetration) which was efficiently delivered to GBM cells, causing cell cycle arrest, apoptosis and downregulation of Notch target genes *Hes-1* and *Hey-L* [229].

Small-molecule inhibitors of the Notch transcriptional complex, such as IMR-1 (prevents MAML1 recruitment to the transcription complex), CB-103 (interferes with assembly of the Notch transcription complex), and NADI-351 (first specific inhibitor of Notch1 transcriptional complex) have exhibited antitumor activity in different xenograft models [230,231,232]. Additionally, antitumor activity of recently discovered small organic molecules (JI051 and JI130) that impair the ability of Hes1 to repress transcription have been demonstrated [233]. These molecules represent an attractive avenue to pursue in the treatment of GBM.

Attempts have been made to inhibit ADAMs, a family of α-secretases that cleave Notch extracellular domain upon ligand binding (Figure 4). Floyd and co-workers reported that an α-secretase inhibitor, INCB3619, decreased proliferation of GBM cell lines as well as GBM stem cell lines mainly through Notch inhibition [234]. Moreover, INCB3619 inhibited tumor growth and prolonged the survival in a human GBM stem cell xenograft model in mice. There is an ongoing phase I trial of INCB7839 (aderbasib), inhibitor of the ADAM 10 and 17 proteases, for children with recurrent or progressive high-grade gliomas (NCT04295759). A group of natural compounds (e.g., thapsigargin) that inhibit sarco-endoplasmic reticulum Ca^2+^-ATPase (SERCA) and affect intracellular trafficking of the Notch receptor causing accumulation of unprocessed Notch1 in the ER/Golgi compartment have emerged as potential therapeutics in cancers associated with *NOTCH1* mutations [235]. In a phase II trial of mipsagargin (a thapsigargin prodrug) in patients with recurrent or progressive GBM (NCT02067156), drug treatment led to disease stabilization in 22% of patients [236].

A number of FDA-approved drugs used to treat other diseases/cancers are able to modulate Notch signaling in different cancer cell lines and could potentially be considered for treatment of GBM. These encompass histone deacetylase (HDAC) inhibitors (e.g., vorinostat) [237], enhancer of zeste homolog 2 (EZH2) inhibitors (which now include FDA-approved drug tazemetostat) [238], a thalidomide derivative lenalidomide [239], and antibiotic quinomycin A [240], most of which have been shown to have an antitumor effect on GBM cell lines in vitro or in vivo [241,242,243]. In addition to blockage of the SHH signaling pathway, ATO downregulates Notch signaling by decreasing the levels of Notch1 and Hes1 proteins having an inhibitory effect on GBM cancer stem-like cells in vitro and in vivo [244]. The other study showed that ATO depletes cancer stem-like cells in GBM and inhibits neurosphere recovery and secondary neurosphere formation by inhibition of the phosphorylation and activation of Akt and STAT3 through blockade of Notch signaling [245]. N-acetylcysteine (NAC) is a sulfhydryl-containing compound, with antioxidant, anti-inflammatory and mucolytic properties, initially used for treating cystic fibrosis, acetaminophen overdose and chronic obstructive lung disease [246]. Recently, Deng and co-workers revealed that NAC could efficiently inhibit Notch2 and its downstream signaling by facilitating Notch 2 degradation through lysosome pathway in GBM cell lines [247]. Moreover, NAC was able to reduce proliferation and induce apoptosis in vitro, and to suppress tumor growth of GBM cells in vivo.

Over the years, a great amount of data regarding the ability of different natural products and their derivatives to modulate Notch signaling and to alter the malignant properties of various cancer cells types have been accumulated (reviewed in [248,249]). They include compounds such as quercetin [250], curcumin [251], butein and its derivative chalcone 8 [252], honokiol [253], resveratrol [254], xanthohumol [255], ginsenoside [256], DATS [257], artemisinin [258], luteolin [259] juglone [260], withaferin A [261] and cucurbitacin B [262]. Most of them display antitumor effect on GBM cells in vitro or in vivo through various mechanisms and by affecting multiple signaling pathways, thereby representing promising alternative or adjuvant therapeutics that need to be further explored to improve outcomes of GBM patients.

The list of developed Notch-targeting approaches for combatting cancer is quite remarkable and is constantly expanding. However, a lot of challenges have to be overcome for successful translation of preclinical results into the clinical setting, including a reduction in drug toxicity associated with Notch inhibition, identification of reliable biomarkers of Notch activity for stratification of patients that would benefit from Notch-targeting therapy, and use of Notch therapies in combination with other agents or conventional chemotherapy or radiotherapy to affect multiple targets and cancer-associated processes simultaneously.

## 6. Targeting TGFβ Signaling Pathway in GBM

TGFβ (Transforming growth factor) is a multifunctional cytokine that plays important roles in the regulation of development and differentiation as well as adult tissue homeostasis (reviewed in [263,264]). In particular, TGFβ is crucial for every step of neural development and is expressed in neurons, astrocytes, and microglia (reviewed in [265]). It is shown that TGFβ and WNT signaling crosstalk controls the growth and size of the developing brain by regulating neural stem cell maintenance and differentiation [266].

The TGFβ family comprises proteins divided into the following: (I) the TGFβ subfamily, which contains TGFβ, activin beta chains, and the protein Nodal, and (II) the bone morphogenetic protein (BMP) subfamily that comprises BMPs, growth differentiation factors (GDFs), and mullerian inhibitory factor (MIF) (reviewed in [267]). In mammals, there are three isoforms of TGFβ (TGFβ1, 2 and 3) [268,269,270]. After activation, TGFβ ligands bind to TGFβ receptor II (TGFβRII), which is a constitutive active kinase that phosphorylates TGFβ receptor I (TGFβRI), thus enabling the transduction of extracellular signal into the cell (Figure 5). In the canonical (Smad-dependent) pathway, TGFβRI activates receptor-regulated (R-) Smad proteins (Smad 2/3), which form transcription regulatory complexes with the Co-Smad (Smad 4) and translocate into the nucleus where they bind to target DNA sequences to regulate the transcription of numerous genes (reviewed in [264,271]). Smad proteins which have an inhibitory function (I-Smads- Smad 6 and 7) suppress phosphorylation and nuclear translocation of Smad 2/3, thus regulating TGFβ through negative feedback mechanism [272] (Figure 5). In addition to the canonical pathway, TGFβ has the ability to activate various signaling pathways. In the non-canonical (Smad-independent) pathway, TGFβ triggers mitogen-activated protein kinases (MAPK) pathways, such as ERK1/ERK2, Jun-N terminal kinase (JNK) and p38 and PI3K kinases (reviewed in [273]).

Under pathological conditions such as neurodegenerative diseases, injury, and cancer, a significant increase in the expression of TGFβ was noticed (reviewed in [275,276,277]). Increased mRNA levels of the TGFβ isoforms were detected in GBM, which correlated with the degree of malignancy and prognosis [278]. TGFβ promotes proliferation of gliomas [279,280], invasion [281], angiogenesis (reviewed in [282]), and maintenance of stemness of patient-derived GSCs via the TGFβ-Sox4-Sox2 pathway [34]. Depending on the stage of malignancy, TGFβ has a dual function. At an early stage of the tumor, TGFβ reduces cell proliferation, promotes cell cycle arrest, induces apoptosis, and induces DNA damage in malignant cells acting as a tumor suppressor (reviewed in [277,283,284]). Later, in an advanced stage of malignancy, TGFβ acts as a tumor promoter directly inducing epithelial–mesenchymal transition (EMT), invasion, and metastasis of malignant cells, or indirectly, by promoting angiogenesis (reviewed in [277,285,286]). TGFβ-Smad activity is elevated in aggressive, highly proliferative gliomas and is related to poor patient prognosis [279].

Recently, many studies highlighted the significance of TGFβ in actively shaping and developing the glioma TME. The TME consists of different cells and extracellular materials surrounding the tumor which are responsible for promoting tumorigenesis through processes such as invasion, metastasis, and resistance to therapy. Studies revealed that various cell types, non-immune and immune cells, are associated with TGFβ activation and secretion during cancer progression. Accumulating evidence support the fact TGFβ has a dual role in glioma (reviewed in [287]). TGFβ secreted by glioma cells maintains tumor growth through inducing immune cells to become immunosuppressive, leading to the lack of an effective immune response against gliomas and formation of a permissive microenvironment. On the other hand, TGFβ produced by immune cells upregulates TGFβRI and TGFβRII on glioma cells and supports tumor progression (reviewed in [288]). Further, TGFβ controls differentiation, angiogenesis, and metabolic reprogramming of stromal cells of the TME during tumorigenesis (reviewed in [289]).

Growing evidence suggests that inhibition of TGFβ signaling could provide novel therapeutic options for treating GBM. Knock-down of TGFβ or TGFβ receptors has been shown to limit migration, invasion, and tumorigenicity of glioma cells [281,290]. Numerous different strategies to target TGFβ signaling pathway in GBM have been established and tested in clinical trials including antisense oligonucleotides, neutralizing antibodies, and kinase inhibitors. One of the approaches tested in clinical trials is usage of antisense oligonucleotides (ASOs). ASOs are specifically designed to bind TGFβ mRNA and inhibit its translation thus decreasing its expression. AP 12009 (Trabedersen) is a synthetic phosphorothioate oligodeoxynucleotide that targets one of the TGFβ isoforms, *TGFβ2* mRNA (reviewed in [291,292]). Hau and co-workers performed in vitro experiments that demonstrated the specificity and efficacy of AP 12009 in patient-derived malignant glioma cells [293]. They also showed that AP 12009 treatment reversed the immunosuppressive effects of tumor-derived TGFβ. In an autologous cytotoxicity assay, where peripheral blood mononuclear cells isolated from glioma patients were activated by IL-2 and co-incubated with the glioma cells from the same patient, treatment with AP 12009 restored their cytotoxic activity against glioma cells. Additionally, three phase I/II studies confirmed that AP 12009 achieved safety and tolerability in patients with recurrent or refractory malignant (high-grade) glioma, anaplastic astrocytoma (WHO grade III) or GBM. Most importantly, in two patients, complete tumor remission was detected, showing promising efficacy of AP 12009 [293]. Lastly, the phase III clinical trial (NCT00761280) using AP 12009 in the treatment of recurrent or refractory anaplastic astrocytoma or secondary GBM, has been terminated due to insufficient recruitment of patients.

Anti-integrin therapy is considered a promising strategy for inhibition of processes involved in the GBM progression [294]. Roth and co-workers demonstrated that cilengitide, a selective integrin inhibitor, reduced phosphorylation of Smad 2 in vivo, confirming that integrins control TGFβ pathway in GBM [294]. A phase III clinical trial CENTRIC (NCT00689221) was conducted to determine overall survival as well as efficacy and safety of cilengitide in combination with standard chemoradiotherapy, compared to standard treatment alone, in newly diagnosed GBM patients with methylated O6-methylguanine-DNA-methyltransferase (*MGMT*) gene promoter. Unfortunately, the clinical trial revealed no improvement in overall survival or progression-free survival of patients [295].

Kinase inhibitors reduce TGFβ kinase activity, thus modulating downstream signaling transduction. In various tumors, including GBM, it is well documented that kinase inhibitors can decrease tumor growth and metastasis, and prevent recurrence and angiogenesis in mouse models [296,297,298,299]. In particular, LY2109761, a TGFβRI kinase inhibitor, reduced the survival of U87 and T98 glioma cell lines, and inhibited migration and angiogenesis. Furthermore, LY2109761 delayed tumor growth in vivo alone or in combination with radiation and TMZ [299]. Galunisertib (LY2157299), a small-molecule inhibitor, reduces a kinase activity of TGFβRI in Smad 2/3 phosphorylation. In a preclinical study, Yingling and co-workers confirmed anti-tumor activity of galunisertib in vitro and in vivo [300]. However, a phase I/II clinical trial showed that galunisertib treatment with TMZ-based chemoradiation had no clinical benefit compared to standard TMZ-based chemoradiation [301]. Further, in another phase II study, treatment by galunisertib in combination with lomustine in patients with recurrent glioma did not improve overall survival relative to placebo with lomustine [302]. Spender and co-workers reported preclinical study regarding antitumor effect of two TGFβRI inhibitors, AZ12601011 and AZ12799734. They demonstrated that AZ12601011 and AZ12799734 were more effective in inhibiting TGFβRI-induced transcription and migration than galunisertib. Additionally, the authors confirmed inhibition of tumor growth and metastasis in vivo using a murine model of breast cancer [303].

Preventing TGFβ signaling transduction is possible by administration of antibodies against ligand or its receptors. To accomplish this objective, several antibodies against TGFβ are developed. Literature data revealed that inhibition of TGFβ signaling by applying TGFβ-neutralizing monoclonal antibody 1D11 increased glioma-associated antigen peptide vaccines efficiency in mice [304]. Further, Hulper and co-workers have demonstrated that 1D11 TGFβ neutralizing antibody can be detected in subcutaneous and intracranial gliomas after intravenous injection [305]. Nevertheless, 1D11 treatment of gliomas had diverse effects on the gliomas in immunocompetent and immunodeficient mice. 1D11 treatment of immunocompetent mice bearing subcutaneous glioma resulted in tumor remission, while the same treatment in immunodeficient mice increased tumor size. Additionally, 1D11 treatment of intracranially implanted gliomas impaired glioma cell invasion in normal brain tissue but did not reduce tumor size [305]. One of the important aspects of treating glioma is the permeability of BBB that often prevents the drugs to reach the target site. Radiolabeled human IgG4 monoclonal antibody that recognizes and neutralize TGFβ, 89Zr-GC1008, showed excellent uptake in patients with recurrent high-grade gliomas with no observed toxicity (NCT01472731). However, 89Zr-GC1008 did not demonstrate an antitumor effect [306].

GSCs, which cause GBM tumor recurrences, are able to inhibit natural killer (NK) cell activity via releasing and activation of TGFβ, thus evading immune attack. It was shown that genetic disruption of TGFβR2 in NK cells results in antitumor activity in vivo [307]. There is an ongoing phase I trial (NCT04991870) conducted to evaluate the best dose, possible benefits and side effects of engineered NK cells, with deleted TGFβRII and glucocorticoid receptor NR3C1, for the treatment of recurrent GBM (reviewed in [308]). Additionally, the trial is aimed to determine overall survival, duration of clinical response, progression-free survival and time to progression.

Using bioactive compounds in combination therapy could be a promising strategy for targeting TGFβ signaling pathways. It was shown that resveratrol, a polyphenolic compound obtained from plants, modulates TGFβ signaling (reviewed in [309]) and exhibits the antitumor effect in various tumors [310,311]. In particular, resveratrol suppresses EMT, migration, invasion, and EMT-generated stem cell-like properties in GBM in vitro via canonical TGFβ signaling, and also inhibits the EMT process in vivo [312].

Taken together, clinical trials data presented above display no sufficient benefit of targeting TGFβ signaling in patients with GBM. The complexity of TGFβ signaling itself as well as interactions with various signaling pathways could explain the discouraging results of clinical studies regarding inhibition of TGFβ signaling in GBM.

## 7. Targeting BMP Signaling Pathway in GBM

The bone morphogenetic proteins (BMPs), the largest part of the TGFβ family, are important regulators of a multitude of processes during embryonic development and homeostasis (reviewed in [313,314,315,316,317]). The body of literature indicates that in the CNS, these proteins have a pleiotropic role during neural stem cell (NSC) proliferation, apoptosis, cell fate decisions and maturation. Furthermore, in the adult neurogenic niches, subventricular and subgranular zones, BMPs are critical for proliferation, maintenance, and survival of NSCs as well as terminal differentiation of newborn neurons (reviewed in [314,315,318,319]), thus profoundly affecting the homeostasis in the adult brain. Interestingly, in contrast to early development, NSCs derived from older animals undergo astrocytic differentiation in response to BMPs (reviewed in [320]).

BMP ligands exert their activities in the cells through both canonical and non-canonical pathways (Figure 6) (reviewed in [317,321]). In the canonical signaling pathway, BMP ligands bind to the cell surface receptors to form a heterotetrameric complex, comprising two dimers of type I and type II serine/threonine kinase receptors. There are three “type I” receptors, type 1A BMP (BMPR-1A or ALK3), type 1B BMP (BMPR-1B or ALK6), and type 1A activin receptor (ActR-1A or ALK2) and three “type II” receptors, type 2 BMP (BMPR-2), type 2 activin (ActR-2A), and type 2B activin receptor (ActR-2B). Following the formation of a heterotetrameric complex, the constitutively active type II receptor trans-phosphorylates the type I receptor and allows phosphorylation of the immediately downstream substrate proteins known as the receptor-regulated Smads (R-Smads): Smad 1, 5, and 8 (Figure 6). R-Smads then associate with the Smad 4, and this complex translocates to the nucleus, where it functions as a transcription factor with coactivators and corepressors to regulate gene expression, as already mentioned in the previous section. In addition, non-canonical (Smad-independent) pathways for BMPs can also lead to regulation of gene expression [321]. It has been found to affect different components including TAK-1, a serine–threonine kinase of the MAPK family, PI3K/Akt, P/kc, Rho-GTPases.

Growing data demonstrate that some BMPs are implicated in different pathologies, including processes related to carcinogenesis, such as angiogenesis and EMT, and driving cancer stem cell resistance to treatments (reviewed in [322,323,324,325,326]). Functional studies revealed that depending on the type of cell, TME, epigenetic background of the patient, or stage of tumor growth, some BMPs can be linked to tumor progression, while others can serve as tumor suppressors (reviewed in [325,327,328]). In glioma, expression analysis of molecular components of the BMP pathway revealed significant differences in the level of expression in tumor tissues that were associated with tumor grade and prognosis pointing to them as novel biomarker with potential important therapeutic implications [329,330,331].

Results of previous in vitro and in vivo studies indicate that BMPs are among the most potent therapeutics in preventing the growth and recurrence of GBM. Exposure of stem-like brain tumor cells to neurosphere-derived BMP7 induced differentiation, reduced stem-like marker expression, self-renewal, and the ability for tumor initiation in mice [332]. Similarly, the BMP7 variant decreased proliferation and induced differentiation of GBM stem-like cells and inhibited angiogenic endothelial cord formation [333]. In the same study, the mouse models with subcutaneously or orthotopically implanted GBM stem-like cells reflected in vitro results [333]. In the study where BMP7 was encapsulated in microspheres in the form which provided an effective release for several weeks, primary tumors showed a growth delay. This effect was correlated with the activation of the BMP canonical pathway [334]. BMP4 is another cytokine whose expression is significantly lower in glioma tumor tissue compared to adjacent normal tissue [330]. Previous studies also showed that BMP4 inhibited cell proliferation and induced apoptosis in GSCs [335] and GBM [336] and initiated GBM-derived stem cell astrocyte differentiation [35]. Liu and co-workers detected decreased expression of BMP4 in the multi-drug resistant (MDR) U251 cell line (U251/TMZ) compared to the parental U251 cells. In line with this result, the overexpression of BMP4 in U251/TMZ cells abolished the MDR both in vitro and in vivo through modulation of Bcl-2 and GDNF [337]. BMP2 also makes GSCs more susceptible to TMZ treatment through destabilization of HIF-1 [338], further suggesting BMPs are promising candidates for GSC-targeting GBM therapy. However, the main obstacle could be that glioma stem cells have mechanisms to avoid BMP-induced differentiation as they express a BMP antagonist Gremlin1 [339].

A recent study on patient-derived GSCs showed that treatment with BMP4 caused downregulation of stem cell marker CD133 expression, induced asymmetric cell division and suppressed self-renewal ability of cells [340]. An extensive study using 40 different human GBM-initiating cell cultures demonstrated an extensive diversity in the inhibitory effect of BMP4 on the tumor growth between cell lines [341]. Responsiveness to BMP4 was correlated with a level of SOX2 and proneural mRNA expression in cells [341].

In the contrast to the aforementioned therapeutic strategy to induce BMP signaling to initiate differentiation of GSCs, most recently, Kaye and co-workers showed that inhibition of BMP signaling could also have potential therapeutic relevance for fighting GBM [342]. In this study, BMP receptor inhibitors, DMH1, JL5, and Ym155 significantly decreased the expression of the inhibitor of differentiation protein 1 (ID1) in GBM cell lines [342]. The members of the ID family of protein, are co-expressed in diverse cell populations of GBM and conditional deletion of *ID1*, *ID2* and *ID3* alleles suppressed the GBM tumor growth and diminished the population of GSCs (reviewed in [343]). Furthermore, Kaye and co-workers demonstrated that spheres formed from GBM cell lines after BMP4 treatment were smaller in size and number. However, treatment with JL5 completely prevented the formation of spheres, suggesting that inhibition of BMP signaling suppressed self-renewal more than its activation [342].

In addition to activation of direct effectors of BMP signaling, such as BMP4 and BMP7, simulation of the BMP pathway is performed using mimic effectors. Recently, the BMP2 protein mimicking peptide GBMP1 has been synthesized by Rampazzo and co-workers and demonstrated to enhance osteogenic differentiation of mesenchymal stem cells and astroglial differentiation of GSCs in vitro [344]. However, further studies are necessary to prove the usage of this drug in clinical studies.

To this day, numerous strategies that enable modulation of the BMP pathway in vivo (starting from upstream of receptors to effector activity) in different cancer and non-cancer-related pathologies have been described in the literature (reviewed in [24,326,345]). However, to the best of our knowledge, there is only one active phase I clinical trial for the treatment of adults with progressive and/or multiple recurrent GBM, in which human recombinant BMP4 is being administered through intratumoral and interstitial convection-enhanced delivery (CED) (NCT02869243) (reviewed in [24]). The reasons for a low number of clinical trials may come from the fact that these tumors are highly heterogeneous, and that BMP signaling regulates a large network of genes and pathways, and only a small portion of those directly relates to the disease pathology or may even have the opposite effect on the clinical outcome.

## 8. Targeting Hippo Signaling Pathway in GBM

The Hippo pathway is an evolutionarily well-conserved signaling pathway involved in tissue development and regeneration that controls organ size by regulating cell proliferation and apoptosis [346]. In physiological conditions, the Hippo pathway suppresses growth, mediates stress-induced apoptosis or regulates cell fate decisions [347]. The Hippo pathway consists of a complex cascade of serine/threonine-protein kinases and its core kinases MST1/2 (Mammalian STE-like protein kinase 1/2) and LATS1/2 (Large tumor suppressor 1/2) [348]. Their roles are to inhibit the transcription cofactors YAP (Yes-Associated Protein 1) and its ortholog TAZ (Transcriptional co-activator with PDZ-binding motif). When the pathway is initiated by activation of MST1/2 associated with SAV1 (Salvatore) they further phosphorylate LATS1/2 and its cofactor MOB1, which in turn phosphorylates the transcription cofactors YAP/TAZ, leading to inhibition of their nuclear translocation [349] (Figure 7).

Phosphorylated YAP/TAZ is sequestrated in the cytoplasm or degraded by the proteasome [350]. When the Hippo pathway is inactivated, the unphosphorylated YAP/TAZ complex is translocated to the nucleus, where it binds the TEAD (TEA domain) transcription factor family. Indeed, YAP/TAZ association with TEAD transcription factors is essential to the control of several targeted genes, such as *MYC* (MYC proto-oncogene bHLH transcription factor), *BIRC5* (baculoviral IAP repeat containing 5), *AXL* (AXL receptor tyrosine kinase), *CTGF* (Connective Tissue Growth Factor) and *CYR61* (Cysteine Rich Angiogenic Inducer 61) involved in cell proliferation and survival [351]. This pathway is regulated by cell–cell contact, cell polarity, and actin cytoskeleton, as well as a wide range of signals, including stress signals, cellular energy status, and mechanical and hormonal signals that act through G-protein-coupled receptors (GPCR) [352].

Since MST1/2 and LATS1/2 core constitute a regulatory part of the Hippo signaling associated with a tumor suppressor effect, the transcriptional cofactors YAP/TAZ associated with TEAD transcription factors represent the terminal effectors of this pathway and play a pro-oncogenic role. YAP/TAZ activation is involved in cell proliferation, mesenchymal transition, invasion, and metastasis, as well as in cancer stem cell maintenance and chemoresistance [353]. Elevated levels and nuclear localization of YAP and in some cases TAZ have been reported in a majority of solid cancers, suggesting widespread deregulation of Hippo signaling in human neoplasia [354]. High levels of nuclear YAP1 immunoreactivity were detected in GBM tissues, and it was shown that silencing of YAP1 in glioma cell lines led to a significant reduction in cell growth [355]. In line with this, a comparison of single-cell RNA-seq datasets from patients with GBM revealed that YAP and TAZ drive a regulatory network associated with the GSC state [356]. It has been shown that YAP/TAZ are required to establish GSC properties in primary cells and required for tumor initiation and maintenance in vivo in different mouse and human GBM models [356]. Further, Bhat and co-workers showed elevated expression of TAZ in GBM tissues and associated TAZ expression with mesenchymal gene signature showing a direct role of TAZ and TEAD in driving the mesenchymal differentiation in malignant glioma [67]. Various studies have also found that hyperactivation of YAP/TAZ is associated with resistance to canonical chemotherapies, radiotherapies, and targeted therapies [357,358]. Current evidence suggests that multiple mechanisms contribute to the deregulation of YAP and TAZ in human cancers, including promoter hypermethylation, mutation, and amplification [354]. Reasonably, YAP/TAZ represents one of the most attractive targets in anticancer therapy as final effectors of the Hippo signaling pathway.

High-throughput screening of approximately 3300 FDA-approved drugs for inhibitors of the transcriptional activity of YAP identified 71 hits including several porphyrin compounds, protoporphyrin IX (PPIX), hematoporphyrin (HP), and verteporfin (VP), which stood out among the top candidates [359]. These compounds disrupt physical interactions between YAP and TEAD, thereby inhibiting YAP transcriptional activity, but more importantly, they are able to cross BBB and accumulate in the brain [360]. Vigneswaran and co-workers showed that VP acts as an inducer of apoptosis of patient-derived GBM cells that successfully suppressed expression of YAP/TAZ transcriptional target genes, and had significant survival benefit in an orthotopic xenograft GBM model [361]. Further, a phase 0 clinical trial reported that liposomal VP was effectively absorbed by GBM cells in patients’ tumor tissue, and VP-treated participants preliminarily showed lower YAP/TAZ protein levels compared with a representative untreated control patient [361]. Further studies are needed to elucidate their efficiency in therapeutic protocols, alone or more likely in combination with established treatment regiments.

Another approved drug that emerged as an interesting therapeutic in GBM is valproic acid (VPA). It has been demonstrated that VPA inhibited glioma cell proliferation, migration, and invasion [362]. Experiments on human GBM cell lines have shown that VPA had a strong antiproliferative effect and also led to a reduction in CD44 expression (a cancer stem cell marker) [363]. Further, it has been shown that CD44 is upregulated in GBM tissue samples and that its depletion blocked glioma cells growth in vitro and in vivo and sensitized GBM cells to cytotoxic drugs in vivo [364]. CD44 functions upstream of the mammalian Hippo signaling pathway, and when CD44 is silenced, it is followed by sustained phosphorylation of MST1/2 and LATS1/2 and, consequently, phosphorylation/inactivation of YAP [364]. There have been several trials of VPA in combination with TZM and radiation for the treatment of brain tumors. Results of a single-group study where newly diagnosed GBM patients started VPA treatment following maximal safe resection (all of them were treated with chemoradiation and TMZ) demonstrated no significant treatment-related toxicity; younger patients (age ≤ 45 years) showed a significantly better overall survival (25 months) versus older patients (8 months) [365]. Limitations of this study include the small sample size, no randomization, and the lack of information regarding other potential prognostic factors. Yuan and co-workers performed a meta-analysis using EMBASE, MEDLINE, ClinicalTrials.gov, and Cochrane Central Register of the Controlled Trials databases to assess the effects of VPA on survival times and GBM recurrence [366]. A total of 1634 patients with a confirmed GBM diagnosis were examined in this meta-analysis, and the authors concluded that GBM patients using VPA showed a relatively better outcome when compared to patients using no antiepileptic drugs or other-antiepileptic drugs [366]. This and similar retrospective studies can be useful as a guide in planning other clinical trials investigating the impact of co-medication with VPA in GBM patients.

Amlodipine, a calcium channel blocker medication used to treat high blood pressure and coronary artery disease, has been shown to provoke actin cytoskeleton remodeling and new assembly of F-actin that, in turn, starts kinase cascade and phosphorylates YAP, leading to its degradation [367]. The authors showed that amlodipine inhibits the survival of GBM cell line LN229 by suppressing YAP/TAZ activities and preventing their accumulation in the cell nucleus [367]. Amlodipine, along with several other antihypertensives, has demonstrated promising preclinical results, but unfortunately, it has not gone into clinical trial yet.

Bazedoxifene (BZA) is a small molecule inhibitor currently used in the clinic as a third-generation selective estrogen receptor modulator to treat postmenopausal osteoporosis [368]. BZA preferentially disrupts the gp130-IL6 receptor complex [369], and a recent study displayed that BZA treatment also accelerated YAP phosphorylation, hypothesizing that there is a cross-talk between IL-6-gp130 and YAP [370]. Wightman and co-workers identified BZA as a candidate therapeutic for GBM, showing its ability to penetrate the BBB and increase the time of survival in an orthotopic syngeneic mouse model [371]. Additionally, as presented by Fu and co-workers, BZA combined with paclitaxel had a stronger ability to suppress YAP signaling and inhibit GBM tumor growth in the orthotopic GBM mouse model [370].

It has been already reported that statins impair the viability of GBM cell lines through TGFβ inhibition [372]. Importantly there is evidence that statins, in particular fluvastatin, could inhibit nuclear localization of YAP and TAZ and, consequently, YAP/TAZ-TEAD-dependent reporter activity [373]. A meta-analysis performed by Xie and co-workers included a total of 2519 patients with GBM, including 430 statin users and 2089 nonstatin users [374]. Analyzed data regarding progression-free survival and overall survival revealed that the use of statins was not associated with prolonged survival of patients with GBM [374]. However, a subgroup analysis showed that the use of statins before diagnosis favors the overall survival of GBM patients, while statin usage after diagnosis might be harmful [374]. Currently, there are two clinical trials evaluating the use of statins: a phase I trial evaluating the safety of fluvastatin and celecoxib (Celebrex) association in gliomas (NCT02115074), and a phase II study assessing the efficacy and safety of atorvastatin in combination with radiotherapy and TMZ in GBM (NCT02029573).

In summary, there are many approved drugs that are considered for repurposed use in GMB, and novel randomized trials are needed for proving their therapeutic efficacy. The general opinion is that repurposed agents are more likely to be combined with current standard regimens, suggesting that they should be considered when planning future trials.

## 9. Targeting Retinoic Acid Signaling Pathway in GBM

Retinoic acid signaling has key roles in vertebrate development [375]. Functions of retinoic acid (RA), a morphogen synthesized from vitamin A [376], in the regulation of different processes, such as fate specification, differentiation, proliferation, apoptosis, immune response, homeostasis, regeneration and maintenance of circadian rhythms have been reported (reviewed in [377,378,379]).

Postnatally, retinoids are derived from carotenoids and retinyl esters. Following ingestion, carotenoids are cleaved into retinal and then reduced to retinol, while retinyl esters are hydrolyzed to retinol. Upon entering into the bloodstream, retinol is bound to retinol-binding protein 4 (RBP4) and this complex enters cells via stimulated-by-retinoic-acid 6 (STRA6) (reviewed in [380]) or by membrane diffusion [381] (Figure 8). Intracellularly retinol is metabolized to RA through a series of oxidation steps (reviewed in [377]). Namely, within cells, retinol is converted to retinaldehyde by cytosolic alcohol dehydrogenases (ADHs) and microsomal retinol dehydrogenases (RDHs). After that, retinaldehyde is oxidized to RA by three aldehyde dehydrogenases ALDH1A1-A3 (reviewed in [380,382]). There are three naturally occurring RA stereoisomers: all-trans retinoic acid (ATRA), 13-cis retinoic acid (13-cis RA) and 9-cis retinoic acid (9-cis RA) (reviewed in [383]). Acting on the producing (autocrine signaling) or the receiving (paracrine signaling) cells, RA enters the nucleus via cellular RA-binding protein 2 (CRABP2) (reviewed in [384]) and interacts with retinoic acid receptor (RAR-RARα, RARβ, RARγ)/retinoid X receptor (RXR-RXRα, RXRβ, RXRγ) heterodimer. Subsequently, this complex interacts with RA-response element (RARE) in the promotor region of RA target genes influencing the transcription of over 500 genes (reviewed in [377]). RA is inactivated by cytochrome P450 family 26 (CYP26) oxidase (reviewed in [382]).

Dysregulation of the RA signaling pathway underlies the etiology of different malignancies, such as leukemias, neuroblastoma, lung, skin, breast, ovarian, prostate, head/neck, pancreatic, liver and bladder cancers, renal cell carcinoma and GBM (reviewed in [391]). On the other hand, RA and its derivatives are used for the treatment of cancers. ATRA is a very effective and curative therapy for patients with acute promyelocytic leukemia and a potential therapeutic agent against oral squamous cell carcinoma and oral dysplasia (reviewed in [383]). Preclinical studies demonstrated the usefulness of 9-cis RA in the prevention of prostate and breast cancers and randomized trial of 13-cis RA showed promising results in the treatment of children with high-risk neuroblastoma (reviewed in [383]). It has been reported that RA inhibits the proliferation of tumor cells, promotes their differentiation, induces apoptosis, and inhibits angiogenesis and metastasis [392].

A series of alterations in the RA signaling pathway was reported in GBMs. Methylation in the *RBP1* gene was identified in *IDH1* and *IDH2* mutant tumors. GBM patients with *RBP1*-unmethylated tumors had decreased median overall survival compared with the patients with *RBP1*-methylated GBMs [393]. Additionally, *CRABP2* was suppressed in GBM through epigenetic silencing [394]. Sanders and co-workers demonstrated increased expression of *ALDH1A2* in GBM compared to the expression detected in low-grade gliomas and upregulation of *ALDH1A2* expression upon GBM recurrence [395]. Campos and co-workers revealed that the expression of *ALDH1A1-3* was decreased in GBMs compared to expression detected in non-tumorous brain tissue [396]. Additionally, *CRABP2* expression was reduced in high-grade gliomas [394] and when comparing primary GBM tissues of short-term and long-term survivors Barbus and co-workers detected that *RBP1* and *CRABP2* expression was higher in GBMs of short-term survivors [397]. Methylation of RARβ was detected in about 70% of GBMs [398]. Literature data showed that glioma cell lines express RARγ [399], while primary cultures of biopsy material from human GBMs expressed RARγ and RXRα [399]. Furthermore, stem-like glioma cells, isolated from primary GBM, expressed RARα. ATRA induced differentiation of these cells, causing antitumorigenic, anti-invasive, antimigratory, antiangiogenic, growth-inhibiting and proapoptotic effects [400]. A block in the retinoid receptor proteasomal degradation pathway and accumulation of sumoylated and high-molecular-weight forms of retinoid receptors that lack transcriptional activity were revealed in glioma stem-like cells [401].

The literature data demonstrated that retinoids might affect the malignant behavior of glioma cells. Bouterfa and co-workers treated glioma cell lines and primary cultures of GBMs with different retinoids (ATRA, 9-cis-RA, 13-cis-RA) and demonstrated that glioma cell lines are generally unresponsive to retinoids, while treatment of primary cultures of GBMs reduced the proliferation and migration of these cells [399]. It has been shown that RA increases the proliferation of GL-15 GBM cells at low concentrations and inhibits proliferation of these cells at high concentration [402]. Additionally, the proliferation of GL-15 GBM cells was inhibited by two structurally related RARγ agonists, CD-437 and CD-2325 [402]. In vitro experiments using GBM cell lines demonstrated that ATRA might induce morphological changes, differentiation, apoptosis, change in the mode of cell migration and reduction in growth, proliferation, invasiveness and adhesion of these cells [403,404,405,406]. Additionally, ATRA increased the asymmetric cell division of GSCs isolated from the U87 GBM cell line [407], induced differentiation and decreased proliferation and invasiveness of U87 cancer stem-like cells [408], induced morphology changes, growth arrest and differentiation of GBM stem-like cells [409] and induced differentiation and apoptosis and reduced proliferation and self-renewal of neurospheres of GBM therapy-resistant cancer stem cells [410]. On the other hand, ATRA treatment of brain tumor stem cells, isolated from the fresh GBM specimen, promoted proliferation and induced differentiation of these cells [411]. Additionally, ATRA has a pro-proliferative and pro-survival effect on stem-like glioma cells mediated by RARα and RARγ [412]. It has been demonstrated that cis-RA treatment of glioma cells may inhibit proliferation and invasiveness and induce differentiation of these cells [399,413,414], while treatment of GSCs with 9-cis RA and 13-cis RA inhibited proliferation and induced their differentiation into neurons, astrocytes and oligodendrocytes [409,415,416]. 13-cis RA showed a significant inhibitory effect on proliferation and clonogenicity of U343 GBM cells [417]. Bexarotene, a RXR agonist, induced morphological changes and differentiation of cultured primary GBM cells and inhibited their neurospheroidal colony formation and migration [418]. Bexarotene or ATRA, alone, reduced tumor size; on the other hand, when treatment was discontinued, the tumor size started to increase [418].

There are literature data about the combined effects of retinoids and other drugs/compounds on the malignant behavior of GBM cells and GSCs. Namely, ATRA in combination with TMZ enhanced TMZ effects on malignant behavior of GBM cells [419], in combination with interferon-gamma (IFN-γ), it induced apoptosis of GBM cell lines [404], and in combination with taxol, paclitaxel or IFN-γ, it induced differentiation, apoptosis and reduction in tumor volume of xenografts of GBM cell lines [420,421,422]. Combined treatment of metformin and 9-cis RA reduced the proliferation rate and increased apoptosis in C6 glioma stem-like cells [423], while 13-cis RA combined with thalidomide delayed the growth of GBM xenografts [424]. Reduced proliferation, invasion and migration of U87 cells, decreased number of colonies of these cells, increased number of apoptotic cells and reduced tumor volumes have been found upon treatment of cells with 6-OH-11-O-hydroxyfenantrene (IIF), an RXR agonist, and pioglitazone, a synthetic PPARγ agonist [425]. Systemic administration of TMZ combined with convection-enhanced delivery (direct intracranial drug infusion technique) of polymeric micellar Am80, a synthetic agonist with high affinity to nuclear RAR, provided longer survival of rats with GBM xenografts compared to controls [390].

The therapeutic benefit of retinoids for therapy of GBM is largely contradictory. Results obtained by Pitz and co-workers demonstrated that there are no significant differences between the survival of GBM patients who were treated with TMZ and cis RA for up to 24 months after surgery and radiation therapy and GBM patients treated for 6 months with TMZ [426]. Additionally, the results of a phase II clinical trial demonstrated that a retinoid combined with TMZ did not increase progression-free survival of GBM patients [155]. Additionally, a phase II trial was conducted to evaluate the effect of concurrent treatment with TMZ and 13-cis RA in combination with conventional radiation therapy in adults with supratentorial GBM, which did not show a survival advantage compared with studies using radiation therapy with TMZ [427]. Furthermore, the phase II trial of fenretinide (4-hydroxyphenyl-retinamide) (NSC 374551), a synthetic derivative of ATRA, in adults with recurrent GBM did not demonstrate clinical efficacy [428]. On the other hand, the combination of 13-cis RA (Accutane) and celecoxib, a COX-2 inhibitor, demonstrated a modest effect on progression-free survival of patients with progressive GBM, but this combination was not more effective than 13-cis RA alone [429]. A phase II evaluation of TMZ and 13-cis RA (NABTC 98–03) revealed a 6-month progression-free survival rate of 32% for patients with GBM [430].

On ClinicalTrial.gov, six studies were found by searching with keywords “retinoid” and “GBM”. Among them, the only one active is focused on evaluating the effects of combined therapy of vorinostat, isotretinoin and TMZ in patients with GBM (NCT00555399). The results of the study NCT00112502 indicated that adding isotretinoin to dose-dense TMZ may be detrimental [155]. Furthermore, there are two on-going studies analyzing the effects of ketoconazole, an inhibitor of the CYP26A1 enzyme. The first study (NCT04869449) analyzes if ketoconazole can enter brain tumors (GBM) at a high enough amount to stop the tumor cells from dividing, while the second (NCT03796273) studies the side effects and how well ketoconazole works before surgery in treating patients with glioma that has come back.

The main limitations of pharmacological applications of RA include poor solubility in aqueous solutions, photosensitivity, rapid metabolism of RA upon intravenous administration, which reduces its efficiency, and side effects after systemic delivery (reviewed in [431]). The results obtained by using ATRA-encapsulated polymeric micelles of a chitosan graft copolymer indicated that encapsulated ATRA is more effective at inhibiting U87 cell migration than free ATRA [432]. Additionally, to stabilize ATRA, Jones and co-workers used a porous poly(1,8-octanediol-co-citrate; POC) wafer which enabled slow release of ATRA leading to differentiation, apoptosis, and inhibition of proliferation of U87 GBM cells [433]. Generally, several strategies for the delivery of RA were used, and the most common strategy used is liposomal or polymeric nanoparticles formed by polyesters, polyimines, polysaccharides and proteins (reviewed in [431]). In conclusions, the use of drug delivery systems that enhance RA solubility, prolong its presence in circulation, and decreased its toxicity might improve its efficiency in cancer treatment, including GBM (reviewed in [431]).

## 10. Conclusions and Future Perspectives

Common genetic alterations in GBM include the loss of the chromosome arm 10q, alterations in tumor suppressor TP53 and tumor suppressor retinoblastoma RB, amplifications of EGFR and PDGFR, and aberrations in RTK/Ras/PI3K signaling pathways, all of which are major known drivers of GBM pathology (reviewed in [11]). Other frequent mutations include alterations in NF1, PTEN, and MDM2 [38,434]. On the other hand, multiple signaling pathways dysregulated in GBM are involved in the promotion of malignant behavior of GBM cells [12]. Their altered activities in GBM are mostly due to changes in the expression rather than mutations in key pathway components. Mutations in components of signaling pathways presented in this paper are not hallmarks of GBM, although some studies linked several mutations to gliomagenesis. In particular, one study detected mutation in *APC* gene, a WNT signaling component, in GBM patient samples [112]. Further, mutations in TGFβ receptor that inactivate the receptor are detected in early stages of malignant glioma tumorigenesis [435]. Additionally, studies have reported mutations of Notch pathway genes in grade II and III gliomas [436]. Unlike low-grade gliomas, Notch mutations in GBM are very scarce. Since no other driver mutations in signaling pathways have been identified so far in GBM, evidently, regulation of these pathways activity depends of other mechanisms, including epigenetic alterations. Sakthikumar and co-workers provided results of whole genome sequencing showing evidence for enrichment of non-coding constraint mutations in GBM-associated genes, as well as in more than 1776 other genes that have not been previously linked to GBM, but may have a functional impact on the disease [437]. This could offer an explanation for differences in the regulation of gene expression and, consequently, changes in signaling pathways activity. Importantly, GBMs frequently evolve and, within a single patient, could display numerous subtypes, various gene expression profiles, transcriptome patterns, and methylation statuses, all features that favor subclonal selection and direct response to therapy. There is always a risk that some important players are not identified, but with the current knowledge, it becomes obvious that the combination of inhibitors of multiple pathways and other therapies should be considered as a future direction for GBM treatment. Here, we summarized recent findings of the progress made in targeting these signaling pathways in GBM. Although numerous studies of the anti-GBM effects of modulators of various signaling pathways gave promising results in in vitro and in vivo models, only a small fraction of them reached the first phases of clinical trials. Additionally, most of the agents that entered clinical trials as monotherapy or in combination with chemo- and radiotherapy displayed poor clinical activity or lack of survival benefit for patients.

Several reasons may explain these disappointing results. One of the reasons lies in the fact that adequate patients’ molecular stratifications are often lacking. It has been demonstrated that individual prognostic factors of each patient, including *MGMT* methylation status, presence of mutant epidermal growth factor variant III (EGFRvIII), the status of signaling pathways activities, baseline performance status, tumor location, and age could influence the success of the trial and final results [438]. The identification of subtype-specific alternations in genes and their expression and related signaling pathways is a crucial step in the discovery and development of new prognostic and therapeutic strategies to target GBM. For example, in search of the subtype-specific prognostic core genes, Park and co-workers showed that in the mesenchymal subtype of GBM, genes were enriched with Wnt/β-catenin-related genes, suggesting that targeting Wnt signaling would be more effective in this subtype of GBM [439]. El-Sehemy and co-workers showed that Norrin, a Wnt ligand that binds FZD4 and activates canonical Wnt/β-catenin signaling, in GSCs with low expression of proneural factor *ASCL1* (Achaete-scute homolog 1), exerts tumor-suppressive effects via Wnt signaling, while in GSCs with high *ASCL1* expression, Norrin acts as an oncogene by promoting Notch signaling in Wnt-independent manner [440]. These results suggest that Wnt should be considered as a therapeutic target exclusively in GBM with low expression of *ASCL1*, while in the GBM subtype with high expression of *ASCL1*, the inhibition of Notch could be a strategy of choice for therapy of GBM. Thus, stratifications of patients based on molecular profiling of their GBMs, including the activity of signaling pathways, would enable the identification of the most relevant targets for each patient.

Several issues need to be addressed for achieving desirable results in GBM therapy, including GBM heterogeneity and plasticity, as well as TME, which contributes to the tumorigenesis and progression of GBM. Today it is accepted that due to the GBM complexity, it is not likely that a single molecular agent could provide a final therapeutic strategy, and combination therapy approaches are likely to perform better in designing novel clinical trials. Based on this, agents that simultaneously target multiple dysregulated pathways might be involved in the development of future therapeutic strategies for GBM. A number of studies have been aimed to identify or develop small-molecule compounds, natural or synthetic, that are able to target multiple signaling pathways simultaneously to combat cancer. Natural compounds with such properties include resveratrol (affects Wnt/β-catenin, Notch and Smad-dependent TGFβ signaling) [134,254,312], DATS (decreases Wnt/β-catenin and Notch) [137,257], honokiol (downregulates Notch and PI3K/Akt/mTOR) [253,441] and garlic-derived Z-ajoene (affects Notch, Wnt and HH) [442]. In our previous work, we found that extracts from *Phlomis fruticosa* L., *Ononis spinosa* L. and *Anthriscus cerefolium* L. plants and bis-Bibenzyls from the Liverwort *Pellia endiviifolia* have anti-GBM activity in vitro [443,444,445,446], and future research of these extracts and compounds should decipher whether the mechanism underlying their antitumor effect involves targeting dysregulated signaling pathways in GBM. It is important to point out that targeting the same signaling pathway in GSCs and GBM tumor cells can affect diverse sets of target genes and, consequently, different cellular processes. Kaye and co-workers recently demonstrated that both activation and suppression of the BMP signaling pathway had a negative effect on tumor sphere growth by affecting different targets [342]. Thus, there is a constant requirement for further laboratory and clinical research to investigate the mechanisms of receptor signaling downstream pathways in GSCs and GBM tumor cells.

In recent years, 3D and 4D model systems for investigation of different aspects of GBM biology have emerged, holding great potential for the assessment of therapeutic responses and personalized drug screening (reviewed in [447]). They include 3D human brain organoids grafted with patient-derived GSCs or GSC spheres [448] and a 4D platform of GBM patient-derived organoids that self-transforms from 3D cell-culture inserts into histological cassettes [449]. Additionally, different in vitro models of the BBB have been developed (reviewed in [450]). These models can be employed to test the antitumor effect of the inhibitors/modulators of signaling pathways and their ability to cross BBB before considering their inclusion in in vivo or clinical studies. Additionally, significant progress has been made in the development of systems for drug delivery, selective disruption of the BBB using high intensity focused ultrasound, and different systems for intratumoral drug delivery in order to achieve therapeutic drug concentration at the site of the tumor (reviewed in [451]). Having in mind that signaling pathways have key roles in the regulation of cell activity, it is important to develop controlled release systems for targeting signaling pathways in tumor cells in order to avoid targeting normal cells and prevent undesirable toxicity. By controlled release systems, fluctuation of drug concentration might be decreased and side effects might be minimized (reviewed in [452]). It has been shown that hydrogels of natural and synthetic polymers enable controlled release of drugs and targeting, as well as protection of drugs from degradation and metabolism, thereby enhancing treatment efficacy and decreasing toxic effects on normal cells (reviewed in [453]). Nanotechnology enables the delivery of agents into tumor tissues, and the development of stimuli-responsive nanocarriers represents a promising strategy for the delivery of agents that target signaling pathways. Researchers have developed stimuli-responsive nanocarriers that can release drugs into tumor tissue in response to different stimuli, such as temperature, pH, and redox [454,455].

Despite successful outcomes of treatment of other aggressive cancers [456,457]), immunotherapy in GBM faced challenges due to the numerous mechanisms of resistance, including the location of the tumor within the brain and the nature of the BBB, as well as the tumor heterogeneity and its immunosuppressive microenvironment (reviewed in [458]). Numerous immunotherapy strategies for GBM treatment have been employed, including antibodies that reeducate tumor macrophages, vaccinations that introduce tumor-specific dendritic cells (DCs), checkpoint molecule inhibition, and modified T-cells and proteins that help T-cells engage directly with tumor cells (reviewed in [459,460,461]). Even though these strategies applied as monotherapy had only limited benefits for patient survival, preclinical studies showed encouraging results. Strategies that might deliver the real medical benefit for GBM patients include concurrent stimulation of the immune response and inhibition of immunosuppressive components or a combination of immune checkpoint blockade (ICB) with chemotherapy, radiotherapy inducing immunogenic cell death or vaccines (reviewed in [459,460,461]). Currently, there are ongoing clinical trials studying combinations of multiple ICBs and combination of ICBs with radiotherapy or vaccination (reviewed in [460]).

Over the years, single-cell RNA sequencing (scRNA-seq) along with other single-cell profiling techniques have become a powerful tool for examining glioma tumors at a resolution of individual cells, providing a comprehensive insight into the glioma biology, intratumoral genetic heterogeneity, cellular lineages, cancer stem cell programs, TME composition, glioma classification, and response to therapies (reviewed in [462,463]). Unraveling the multiple layers of complexity that characterize GBM will ultimately lead to the identification of novel, more efficient targeted therapies. For example, pathway-based classification by using multiple datasets from GBM scRNA-seq and bulk tumors identified four tumor cell states and GBM subtypes, of which the mitochondrial GBM subtype exhibited significant sensitivity to inhibitors of oxidative phosphorylation, suggesting that GBM patients with this subtype could benefit from targeted metabolic therapy [57]. scRNA-seq profiling of individual GBMs by Neftel and co-workers uncovered that four different malignant cellular states, NPC-like, OPC-like, AC-like, and MES-like, are shared across the GBM subtypes, and that each GBM subtype displays an abundance of distinct cellular states [49,464]. All four GBM cellular states display a proliferation signature, and three of them are able to propagate tumors, with the AC-like state showing decreased potential for tumor initiation. GSCs that may exhibit multiple cellular states, which may also interconvert, pose a great challenge for targeted eradication of GSCs [464]. Suva and Tirosh proposed induction of an AC-like state as a potentially relevant approach in differentiation therapy of IDH-mutant and H3K27M gliomas [464].

In conclusion, although myriad agents targeting signaling pathways dysregulated in cancer have been identified, there is always a need for finding novel drugs or improving the existing drugs in terms of their efficacy and safety in anticancer therapy. Investigating novel or better predictive biomarkers, improving patient stratification, developing of computational methods to accurately predict the response of different parts of the tumor to a given therapy, and decreasing drug toxicities by designing more selective drugs and combinatory regimens that affect multiple targets and processes including proliferation, tumor angiogenesis, and invasiveness, are required to overcome the challenges of signaling pathways-targeting therapies in cancer and in GBM in particular. In the light of current knowledge, the optimal approach to target GBM and control tumor recurrence might be to combine modulators of dysregulated signaling pathways with conventional therapies and immunotherapies.

## Figures and Tables

**Figure 1 cells-11-02530-f001:**
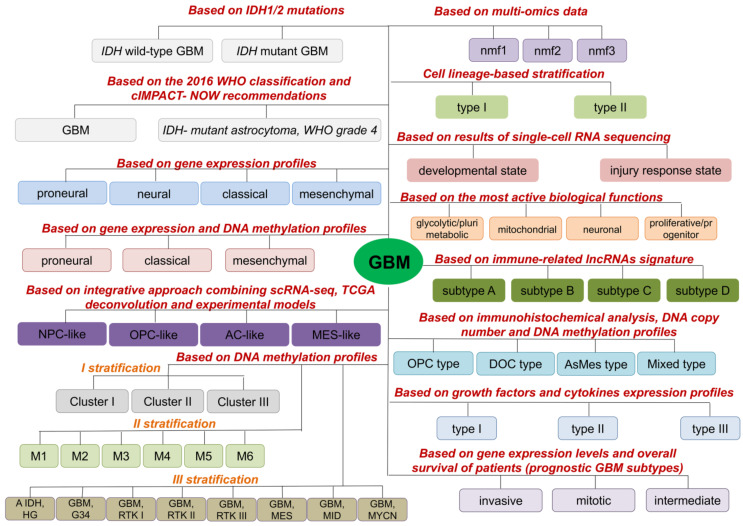
Stratification of GBM into subtypes. This summary is based on the previously reported publications listed in the main text and in Appendix A.

**Figure 2 cells-11-02530-f002:**
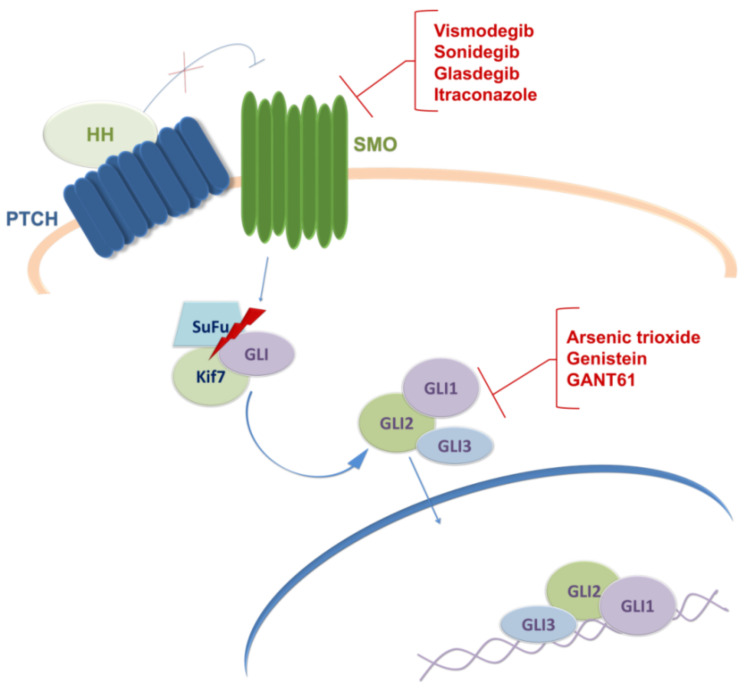
The canonical activation of HH pathway and its pharmaceutical inhibitors. The activation of pathway occurs when HH ligand binds to PTCH at the cell membrane. In response to this binding, PTCH no longer inhibits SMO and initiates the downstream signaling, causing rapid dissociation of the SuFu–GLI complex and thus allowing GLI to enter the nucleus and regulate transcription of target genes. Proven pharmacological inhibitors that target SHH signaling components (SMO receptor or GLI transcription factors) are presented in red. References are included in the main text.

**Figure 4 cells-11-02530-f004:**
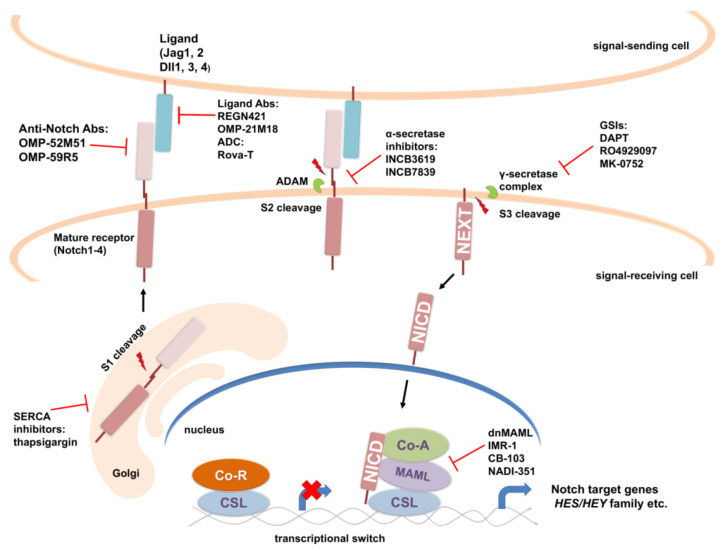
Scheme of a canonical Notch signaling pathway and therapeutic targets. During transport through endoplasmic reticulum and Golgi apparatus, Notch precursor is glycosylated, cleaved into a heterodimer (S1 cleavage) and transported to the cell membrane. Binding with Notch ligand induces second cleavage (S2 cleavage) by a member of ADAM family of proteases, leaving membrane-bound Notch extracellular truncation (NEXT) fragment. NEXT is subsequently cleaved by γ-secretase complex (S3 cleavage) releasing the active form of the Notch receptor, Notch intracellular domain (NICD), which can translocate to the nucleus, where it activates transcription of Notch target genes by forming transcriptional complex with DNA-binding protein CSL (also known as CBF1/in mammals, Suppressor of Hairless in *Drosophila*, and LAG-1 in *C. elegans*) and MAML, which further recruits other transcriptional coactivators (Co-A). Classes of inhibitors and antibodies (Abs, ADC) that target Notch pathway components are indicated. References are included in the main text.

**Figure 5 cells-11-02530-f005:**
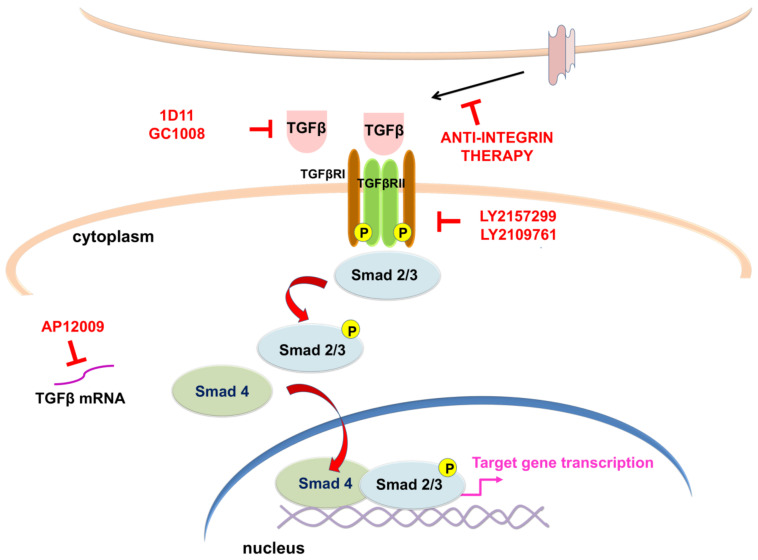
Schematic representation of the canonical (Smad-dependent) TGFβ pathway and TGFβ signaling therapies which have undergone clinical trials in GBM. TGFβ binding to TGFβ receptors II (TGFβRII) results in phosphorylation and activation of TGFβ receptors I (TGFβRI), phosphorylation of Smad 2/3, which interact with Smad 4 and form a complex that translocates into the nucleus to activate target genes. Antisense oligonucleotides (AP12009), anti-integrins, kinase inhibitors (LY2157299, LY2109761) or neutralizing antibodies (1D11, GC1008) used for targeting TGFβ signaling pathway are presented in red. Based on [274] and references included in the main text.

**Figure 6 cells-11-02530-f006:**
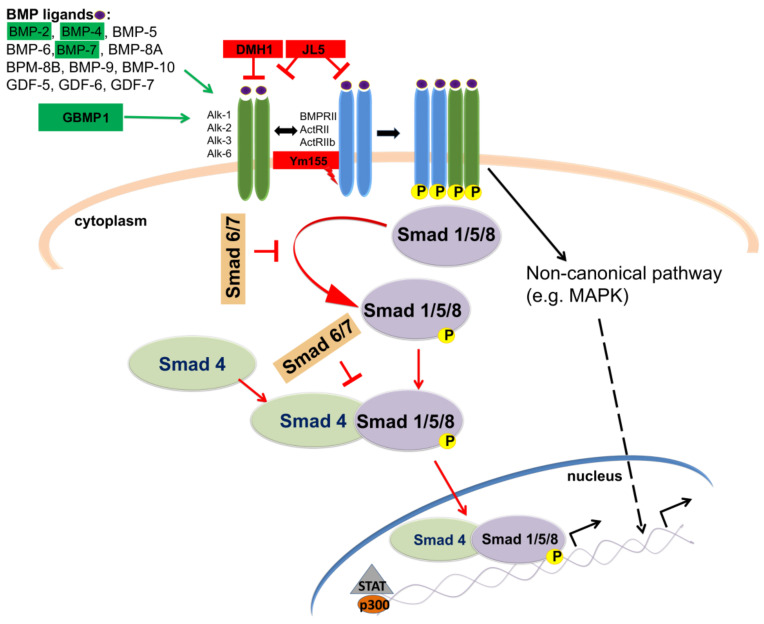
Overview of the canonical BMP cell signaling pathway and molecules investigated for potential therapeutic intervention in GBM. In canonical pathway, various BMP ligands binds to two receptor types (type I and type II) to form a heterotetrameric complex, which then binds to and phosphorylates the receptor-activated Smad 1, Smad 5 and Smad 8. Extracellular inhibitors of BMPs including Noggin, Chordin, and Gremlin inhibit activity of this signaling pathway. Activated Smads (Smad 1, 5, and 8) form complexes with Smad 4, enter the cell nucleus and in combination with co-binding partners, such as p300 or STAT, act as transcription factors and activate multiple gene expression. BMP ligands that activate BMP signaling in the GBM and GSCs are in the green boxes. Receptor inhibitors that suppress BMP signaling in the GBM and GSCs are in the red boxes. DMH1 targets BMP type 1 receptors, JL5 inhibits both the type 1 and type 2 BMP receptors. Ym155 does not bind to the BMP receptors but induces the degradation of BMPR2. References are included in the main text.

**Figure 7 cells-11-02530-f007:**
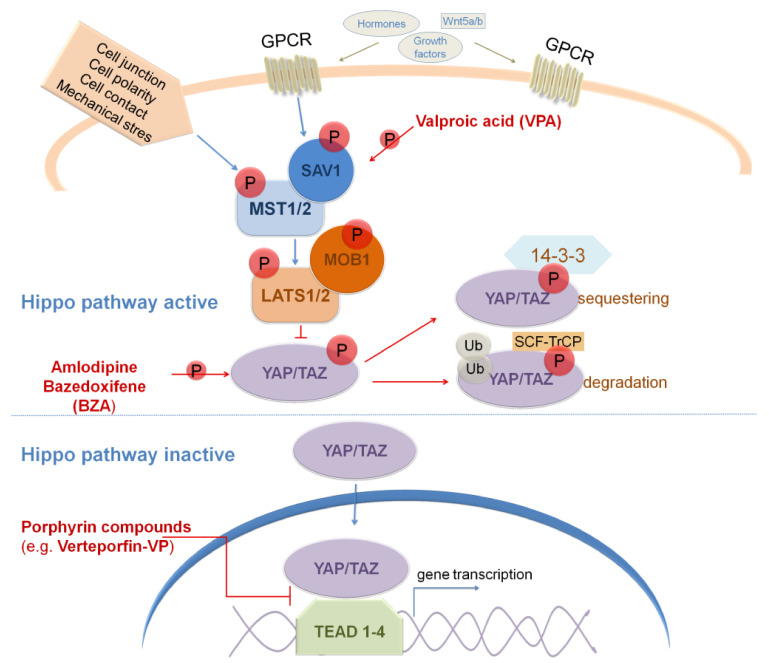
The canonical Hippo pathway and its pharmaceutical inhibitors. Various extracellular signals including mechanical stress, cellular contact, hormones and growth factors activates Hippo signaling cascades that through serial phosphorylations involving block of kinases inhibits nuclear translocation of transcriptional co-activator YAP/TAZ and consequently their involvement in the regulation of gene transcription. When Hippo pathway is inactive, YAP/TAZ translocates to the nucleus, associates with TEAD family of transcription factors and participates in the regulation of target genes expression. Inhibitors that target important pathway components are indicated in red. References are included in the main text.

**Figure 8 cells-11-02530-f008:**
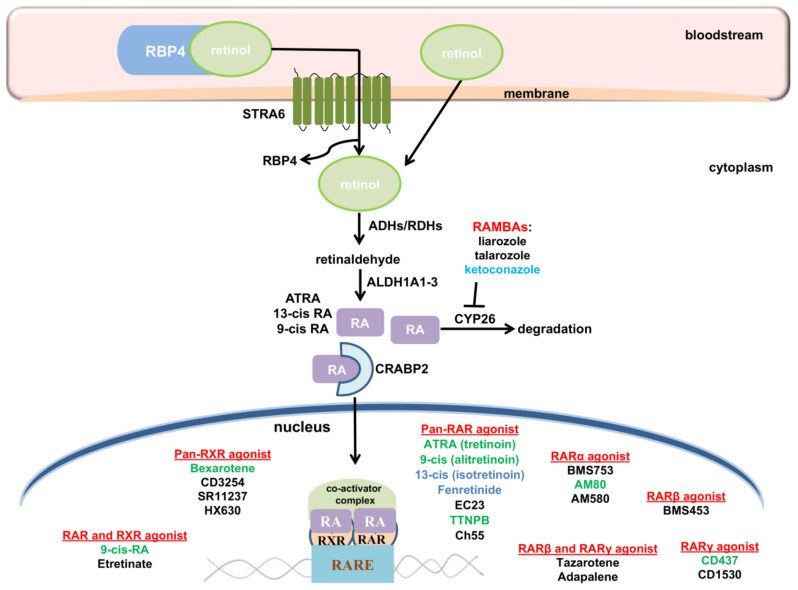
Overview of RA signaling pathway and RAR/RXR agonists and RAMBAs. In the bloodstream, retinol forms complex with RBP4 and enters the cells via STRA6 or by membrane diffusion. Within cells, retinol is converted to retinaldehyde by ADHs and RDHs and subsequently oxidized to RA by ALDH1A1-3. Three naturally occurring RA stereoisomers are ATRA, 13-cis RA and 9-cis RA. RA enters the nucleus via CRABP2, where it interacts with RAR/RXR heterodimer forming a complex that binds to RARE in the promoter regions of RA target genes. RA is inactivated by CYP26 oxidase (modified based on [380]). Summary of inhibitors of the CYP26A1 enzyme (RAMBAs) and RARs/RXRs agonists is made based on [383] and https://resources.tocris.com/pdfs/literature/reviews/retinoid-receptors-review-2019-web.pdf (accessed on 23 May 2022), respectively. RAR and RXR agonists and RAMBAs used for treatment of GBM cells in vitro or in vivo are presented in green letters, and RAR and RXR agonists and RAMBAs used in clinical trials for GBMs are presented in blue letters, while RAR and RXR agonists and RAMBAs not yet used in treatment of GBM are presented in black letters (based on results of previously reported publications included in the main text and results obtained by [385,386,387,388,389,390]).

## Data Availability

Not applicable.

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
