# Peer review of "Current Opportunities for Targeting Dysregulated Neurodevelopmental Signaling Pathways in Glioblastoma"

_cells, 2022, doi:10.3390/cells11162530_

Round 1

Reviewer 1 Report

This is a well written review. 

My most important question would be why authors selected SHH, WNT, NOTCH, TGF-beta/BMP, Hippo and retinoid acid-associated pathways for their review. These are connected to glioblastoma, however other as EGFR, PDGFR, Akt/PI3K/mTOR, cell cycle associated ones (CDK4/6, CDKN2A/B) are possibly more important, more prevalent, and have been addressed by a variety of therapeutic approaches. They can not be neglected in such an review. 

Further, e.g. TGF-beta has it’s major role in modulating the the tumour immune-microenvironment and not as much in the modulation of pathways that wight influence e.g. proliferation and/or invasion, angiogenesis etc. Authors should therefore carefully review their selection and either include all relevant pathways or focus on specific pathophysiological areas as, e.g., shaping of proliferation/apoptosis, angiogenesis, immune-modulation, or whatever they decide to choose as named in lines 51 ff, however with a clear outline that only these sub-topics of the whole landscape are addressed here.

Figure 1: I like the figure, however I am missing classes derived from the EPIC/850K approach (Heidelberg classifier) wich is one of the main classification systems for WHO 2021 and incudes classes as RTK I, RTK II etc.

In line 125 ff, authors claim that the pro neural to mesenchymal with can be induced by NFkapaB. This is correct, however many other inducers have been described which should be added.

The reviewed parts are fine, with an in-depth discussion of the literature.

Reviewer 2 Report

Glioblastoma (GBM) is a fatal malignancy, with a median survival of less than 2 years with current treatments. Some of the molecular alterations in GBM include EGFR, PDGFRA, CDK4, MDM2 amplification, loss of PTEN, NF1, p16INK4 and TERT promoter mutations. The authors of this manuscript have reviewed dysregulated signaling pathways in glioblastoma.  Among several pathways, this review is focused on Wnt/beta-catenin, Notch, Sonic Hedgehog, Bone morphogenetic protein (BMP) and transforming growth factor Beta (TGF-B), Hippo and retinoic acid pathways. This review is exhaustive and gives the feeling of reading a thesis on these pathways. The intention of the authors on advancing therapies for glioblastoma is appreciated. However, there are concerns that need to be addressed before being accepted for publication.

11)     There are no driver mutations associated with the pathways reviewed here. Do the authors agree? Activation or inhibition of the pathways using small molecular inhibitors or antibodies are easily overcome by driver mutations upstream and early in gliomagenesis. This aspect should be discussed in the manuscript.

22)     Wnt, BMP, TGF-B, Notch are all pathways essential for neural development. How is it going to be possible to systemically target these pathways without impacting healthy cells? Do some of the recent sc-RNA-seq studies by Dr. Mario Suva help in cell-specific targeting of the pathways? A seminal paper on glioblastoma classification and plasticity is missing in the manuscript: (https://pubmed.ncbi.nlm.nih.gov/31327527/).

33)     What % of GBM patients have mutations in Wnt/beta-catenin pathway and Smo/gli pathway?

44)     PTEN, AKT, mTOR, c-myc signaling in GBM is equally relevant and important in GBM. What is the rationale for choosing the pathways in this review?

55)     Isn’t it important to identify how the dysregulation of the pathways reviewed changes during glioma development? The authors should discuss this challenge.

66)     Some important studies of the Wnt pathway in gliomas and GBM have been ignored. Griveau et al. Cancer Cell. 2018 (https://pubmed.ncbi.nlm.nih.gov/29681511/), Huang et al. Science Translational Medicine. 2020 (https://pubmed.ncbi.nlm.nih.gov/32102932/). These papers show evidence for translational potential when Wnt pathway inhibitors are combined with chemotherapy.  Focusing on translational studies is more important as compared to studies that solely focus on proliferation in vitro.

77)     The last paragraph on the Wnt pathway is unacceptable. The paragraph starts with talking about Wnt signlaing in GBM. It finishes with a general statement that Wnt signaling’s role in BBB permeability, MDR etc… hold potential for more efficient drug delivery to CNS and more favorable disease outcomes. Wnt pathway inhibition in GBM is complex. Wnt and GPR124 (Wnt receptor) agonists can repair BBB and normalize vessels in GBM (https://www.science.org/doi/10.1126/science.abm4459). Loss of Wnt7a/b function blunts the angiogenic response to hypoxia, resulting in severe white matter damage (Chavali et al. Neuron. 2020) (https://pubmed.ncbi.nlm.nih.gov/33086038/). Endothelial Wnt/Beta-catenin signaling inhibits glioma angiogenesis and normalizes tumor blood vessels by inducing PDGF-B expression. (https://pubmed.ncbi.nlm.nih.gov/22908324/). The challenge of inhibiting Wnt pathway in GBM should be addressed using the above references.

88)     In this era of immunotherapy, the authors have not mentioned a single line about combinatorial targeting of dysregulated pathways with immunotherapy. Targeted therapy has largely failed in GBM, and it is essential to combine it with immunotherapy. A few sentences on this must be added in the manuscript.

99)     Lineage based classification of GBMs as type I and type II performed by Dr. Luis Parada’s group should be cited (https://pubmed.ncbi.nlm.nih.gov/32649888/) as it is relevant for Figure 1. Also with respect to Figure 1 Hubert and Lathia. Nature Cancer. 2021 (https://www.nature.com/articles/s43018-021-00176-x) should be cited and discussed. Although earlier papers of these authors are cited, it is important to cite recent developments for the reader.

110)  The term IDH-mutant GBM has been discontinued upon consensus. IDH-mutant glioblastoma is now referred to as IDH-mutant astrocytoma, WHO grade 4 (EANO guidelines on the diagnosis and treatment of diffuse gliomas of adulthood-Weller et al. NRCO. 2021- https://pubmed.ncbi.nlm.nih.gov/33293629/

Round 2

Reviewer 1 Report

The authors have picked up all my comments and have solved them sufficiently. 

Author Response

Thank you very much for your answer.

Reviewer 2 Report

Since the manuscript's title is now "Current opportunities for targeting dysregulated neurodevelopmental signaling pathways in glioblastoma" 

I'd like to suggest the authors to cite this relevant paper by Dr. Petrecca's group and briefly explain the results in a couple of sentences:

Couturier, C.P., Ayyadhury, S., Le, P.U. et al. Single-cell RNA-seq reveals that glioblastoma recapitulates a normal neurodevelopmental hierarchyNat Commun 11, 3406 (2020). https://doi.org/10.1038/s41467-020-17186-5

The manuscript needs to be proof read since there are a lot of unnecessarily hyphenated words in response to reviewer 2. 

I appreciate that the authors have addressed all my other concerns satisfactorily. The paper can be accepted after these minor additions. 

Author Response

Point 1: Since the manuscript's title is now "Current opportunities for targeting dysregulated neurodevelopmental signaling pathways in glioblastoma" I'd like to suggest the authors to cite this relevant paper by Dr. Petrecca's group and briefly explain the results in a couple of sentences:

Couturier, C.P., Ayyadhury, S., Le, P.U. et al. Single-cell RNA-seq reveals that glioblastoma recapitulates a normal neurodevelopmental hierarchyNat Commun 11, 3406 (2020). https://doi.org/10.1038/s41467-020-17186-5

Response 1: We appreciate very much your comment and think the suggested paper would be a valuable addition to the Manuscript. Accordingly, we added the following text in the Section 1. Glioblastoma:

“Comparison of the lineage hierarchy of the developing human brain to the transcriptome of GBM cells and GSCs derived from IDH-mutant astrocytoma, WHO grade 4 revealed that this type of brain tumor develops along neurodevelopmental gene programs encompassing a rapidly dividing progenitor population [29]. IDH-mutant astrocytoma, WHO grade 4 is hierarchically organized into three cell lineages that correspond to three normal neural lineages, astrocytic, neuronal, and oligodendrocytic, with progenitor cancer cells at its apex [29].”

Point 2: The manuscript needs to be proof read since there are a lot of unnecessarily hyphenated words in response to reviewer 2. I appreciate that the authors have addressed all my other concerns satisfactorily. The paper can be accepted after these minor additions. 

Response 2:  We have carefully proof read the Manuscript to correct unnecessarily hyphenated words. However, words with unnecessary hyphens are present only in the Response to Reviewer 2. They are correctly written in the Manuscript itself and we believe that it happened when we copied the text from the Manuscript and pasted in the Response to the Reviewer 2.